# Identifying and characterizing SCRaMbLEd synthetic yeast using ReSCuES

Zhouqing Luo [1,2], Lihui Wang [2], Yun Wang[3,4], Weimin Zhang[2], Yakun Guo[2], Yue Shen[3,4,5], Linghuo Jiang[6], Qingyu Wu[2], Chong Zhang[7], Yizhi Cai [5,8] & Junbiao Dai [1,2]

SCRaMbLE is a novel system implemented in the synthetic yeast genome, enabling massive chromosome rearrangements to produce strains with a large genotypic diversity upon induction. Here we describe a reporter of SCRaMbLEd cells using efficient selection, termed ReSCuES, based on a loxP-mediated switch of two auxotrophic markers. We show that all randomly isolated clones contained rearrangements within the synthetic chromosome, demonstrating high efficiency of selection. Using ReSCuES, we illustrate the ability of SCRaMbLE to generate strains with increased tolerance to several stress factors, such as ethanol, heat and acetic acid. Furthermore, by analyzing the tolerant strains, we are able to identify *ACE2*, a transcription factor required for septum destruction after cytokinesis, as a negative regulator of ethanol tolerance. Collectively, this work not only establishes a generic platform to rapidly identify strains of interest by SCRaMbLE, but also provides methods to dissect the underlying mechanisms of resistance.

[1] Center for Synthetic Biology Engineering Research, Shenzhen Institutes of Advanced Technology, Chinese Academy of Sciences, Shenzhen 518055, China. [2] Key Laboratory of Industrial Biocatalysis (Ministry of Education) and Center for Synthetic and Systems Biology, School of Life Sciences, Tsinghua University, Beijing 100084, China. [3] China National GeneBank, BGI-Shenzhen, Shenzhen 518083, China. [4] BGI-Shenzhen, Shenzhen 518083, China. [5] School of Biological Sciences, The King's Buildings, University of Edinburgh, Edinburgh EH9 3BF, UK. [6] The Laboratory for Yeast Molecular and Cell Biology, School of Agricultural Engineering and Food Science, Shandong University of Technology, Zibo 255000, China. [7] Department of Chemical Engineering, Tsinghua University, Beijing 100084, China. [8] Manchester Institute of Biotechnology, University of Manchester, Manchester M1 7DN, UK  Correspondence and requests for materials should be addressed to Y.C. (email: yizhi.cai@manchester.ac.uk) or to J.D. (email: junbiao.dai@siat.ac.cn)

In the past decade, substantial progress has been made in the field of genome biology. Firstly, thanks to technology breakthroughs in high-throughput sequencing, the ability to read the genome of an organism has improved enormously. Currently, we are able to read over 35 petabases (approximately 10,000,000 human genomes) each year worldwide[1]. This wealth of sequencing data enables us to discover many novel functional enzymes and pathways, expediting the de novo construction of metabolic pathways in heterologous hosts[2,3]. Secondly, the development of CRISPR-Cas9-mediated genome editing in recent years[4–7] has greatly enhanced our capacity to make targeted changes at one or several specific genomic locus, enabling us to efficiently engineer biology. Finally, facilitated by advances in de novo DNA synthesis technology, an entire genome can be engineered and resynthesized according to arbitrary sets of design principles to accelerate future manipulation. Several synthetic genomes have been chemically synthesized, ranging in length from several kilobase pairs to over one million base pairs[8–11]. An international group of researchers and students have worked together to build the first eukaryotic genome, Sc2.0 (www.syntheticyeast.org). One chromosome arm and six chromosomes have been finished[12–20].

Having the ability to design and construct a new genome enables scientists to incorporate systematic changes throughout the genome. For example, watermarks were introduced at several locations within the synthetic *Mycoplasma genitalium* genome and PCRtags were employed in the Sc2.0 genome, allowing quick identification of presence of the synthetic DNA[9,16]. Seven codons were eliminated to form a "new" *Escherichia coli* genome "rE.coli-57"[21]. More dramatically, a system named Synthetic Chromosome Rearrangement and Modification by LoxP-mediated Evolution (SCRaMbLE) has been implemented in the Sc2.0 genome by introducing thousands of symmetrical loxP sites (loxPsym). Recombination between the loxPsym sites can, theoretically, lead to an enormous number of genome rearrangements such as deletions and inversions, and produce strains with different properties[19,22].

SCRaMbLE has been proposed as a powerful strategy for numerous applications in analyzing genome structure, content and function, with the potential to accelerate evolution and generate conditional minimal genomes[22]. Previous work demonstrated the ability of SCRaMbLE to generate phenotypic diversity, producing strains with heterogeneous growth rates[19], and to delete unessential genomic regions flanked by specific loxPsym sites[20,23]. However, to repurpose SCRaMbLE for a broader range of applications, several limitations have to be overcome. First, although the daughter cell-specific and chemically controlled activation of Cre results in near-zero background, the mother cells, for example, survived after SCRaMbLE. Additionally, there are SCRaMbLE "escapers", which survived due to the mutated Cre gene. These un-scrambled cells can interfere with the identification of strains with rearranged genomes. Second, previous studies have relied on colony size or the deletion of auxotrophic genes to select for SCRaMbLEd strains. Due to variation of growth rate among different clones, however, a small colony size is not a necessary indicator of the presence of rearrangement, and vice versa. While using a particular auxotrophic marker could potentially solve this problem, these markers are not always available. In addition, the loss of these markers is not easy to identify, which usually requires screening millions of clones. Finally, and most importantly, these methods can only select for strains with large variations in growth rate, which is not suitable, for example, to identify strains with a minimal genome but equal fitness to that of the wild type.

In this study, ReSCuES has been developed to efficiently identify SCRaMbLEd colonies in strains containing synXII, the largest synthetic chromosome in *Saccharomyces cerevisiae*. The system utilizes the orthogonality of loxP and loxPsym sequences by designing a cassette of dual auxotrophic marker genes flanked by two inward pointing loxP sites. Only in cells with an active Cre recombinase, the cassette will be flipped, resulting in a switch of "on" and "off" states of the two markers, respectively. Using ReSCuES, we find among five randomly isolated clones, each contains at least three rearrangements in synXII. The SCRaMbLEd synXII are sequencing reconstructed and confirmed, revealing several interesting features. Next, ReSCuES is applied to isolate yeast strains with increased tolerance to several stress

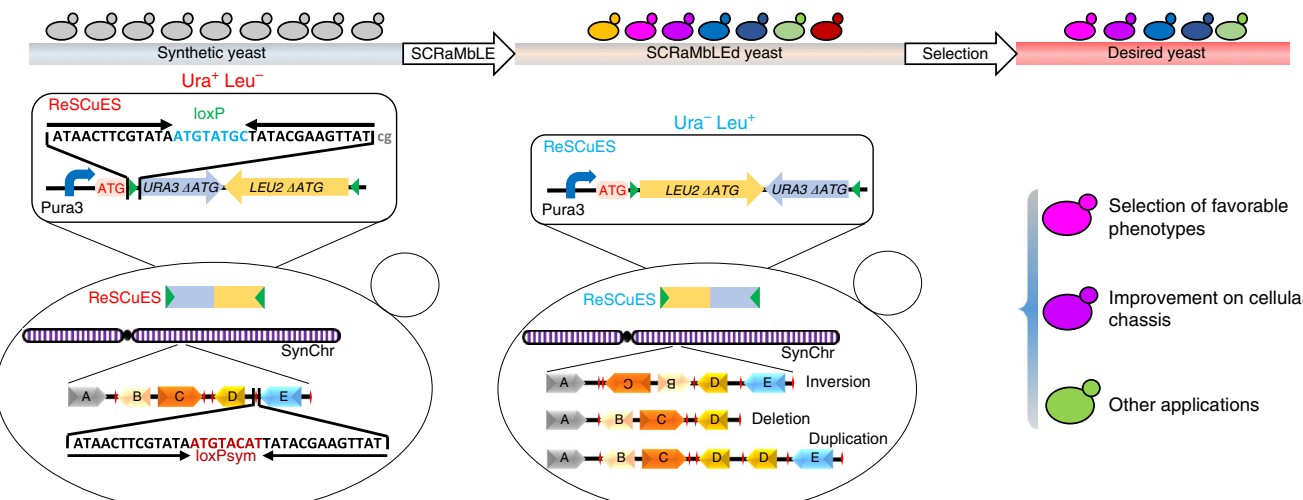

**Fig. 1** Schematic overview of the workflow using ReSCuES and SCRaMbLE. On the left are a population of genetically identical cells containing ReSCuES and one or more synthetic chromosomes (Synthetic yeast). The composition of ReSCuES is illustrated in the rectangle. Pura3 represents the native *URA3* promoter. ReSCuES could present in the yeast cells either in an episomal plasmid or integrated at a chromosome locus, rendering the cells as Ura⁺ Leu⁻. SynChr, synthetic chromosome. Green triangle represents the loxP site. Red diamond represents the symmetric loxP site (loxPsym). In the middle are the SCRaMbLEd yeast strains selected by ReSCuES. Recombination between the two loxP sites inverts the markers, producing Ura⁻ Leu⁺ clones. Simultaneously, rearrangements happen in SynChr, leading to inversion, deletion, and duplication as depictured. In this population, the genome of each cell might be different, which is represented by different colors. On the right are the desired yeast strains after applying certain selection pressures, which could be strains with favorable phenotypes, improvement on cellular chassis and other applications

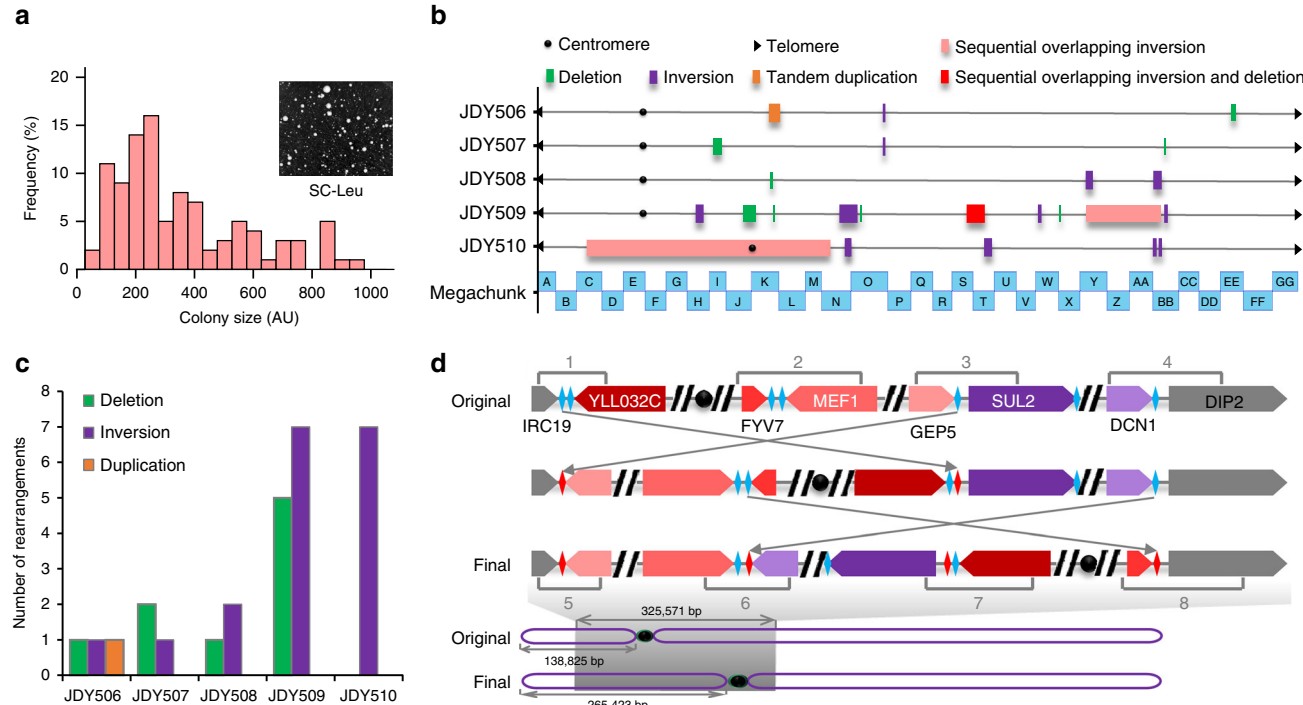

**Fig. 2** Selection and characterization of SCRaMbLEd strains with ReSCuES. **a** Distribution of sizes of SCRaMbLEd synXII clones with ReSCuES integrated at *HO* locus. The colony sizes of 100 Leu$^+$ colonies were determined by ImageJ. AU, arbitrary units. **b** Rearrangements of synXII in five randomly selected colonies. Different types of rearrangements were labeled by different color. The relative size and position were indicated using the synthetic megachunks as reference. The detailed information of each rearrangement was showed in Supplementary Figs. 3–7. **c** The numbers of different rearrangements identified in the five strains. **d** The largest inversion (the shadow area) was found in JDY510 that altered the relative location of centromere. The potential recombination events producing the rearrangement were illustrated. The number indicates the PCR amplicons used to verify the rearrangement

factors, including ethanol, heat, and acetic acid. Using one of the ethanol-tolerant strains as an example, whole-genome sequencing, backcrossing and PCRtag analysis are able to identify the genomic rearrangements on synXII linked to the phenotype. Further analysis reveal that the expression of a key transcriptional regulator is diminished due to the removal of 3′ UTR through a loxPsym-mediated rearrangement, leading to increased ethanol tolerance.

## Results

### Design of a reporter system to enrich the SCRaMbLEd strains.
To facilitate the efficient identification of SCRaMbLEd clones, ReSCuES was designed and constructed based on an alternating on/off switch of two auxotrophic markers, *URA3* and *LEU2* (Fig. 1). Open-reading frames of both *URA3* and *LEU2* without ATG start codon were positioned adjacent to each other and in a convergent direction to make the "*URA3-LEU2*" module. The module was flanked by two opposite loxP sites, which are orthogonal to the loxPsym sites[24]. The *URA3* promoter plus ATG codon was inserted upstream of the "*URA3-LEU2*" module, allowing the expression of a functional Ura3p (Fig. 1, Synthetic yeast). Activating ReSCuES by Cre-mediated recombination between the two loxP sites will result in the inversion of the "*URA3-LEU2*" module, which will subsequently turn on the expression of *LEU2* gene while, simultaneously, shutting down *URA3* expression. Consequently, only strains with activated ReSCuES can survive in media lacking leucine (Fig. 1, SCRaMbLEd yeast). Therefore, ReSCuES can be used as a faithful reporter of Cre activity within a cell.

ReSCuES can be delivered into the cell either as a replication-competent episome or integrated at a genomic location such as the commonly used *HO* locus. Since no recombination between wild type loxP and loxPsym sites will happen either in vivo or in vitro[24], ReSCuES can reside stably in a designated location away from synthetic chromosomes. Given the presence of hundreds of loxPsym sites within synXII, it is highly likely that there are also other rearrangements among the loxPsym sites when ReSCuES is activated. Therefore, a population of SCRaMbLEd new strains can be selected on leucine-deficient media, avoiding laborious screens of thousands of clones (Fig. 1, SCRaMbLEd yeast). These strains can then be subjected to various applications, such as identifying a desired phenotype or generating a more efficient metabolic chassis (Fig. 1, Desired yeast).

### Isolate SCRaMbLEd strains using ReSCuES.
In order to test whether ReSCuES functions as expected, we integrated the reporter at the *HO* locus of BY4741, and then introduced the *Pscw11*-Cre-EBD plasmid (*CEN, HIS3*), or an empty plasmid as the negative control. No colonies grew on synthetic complete medium lacking leucine (SC-Leu) without the Cre-EBD plasmid (Supplementary Fig. 1a), indicating that ReSCuES by itself can stably exist in the genome. In the presence of Cre recombinase but without estradiol induction, a few Leu$^+$ colonies could, very occasionally, be found after long incubation (24 h), indicating a very low leaky expression of Cre-EBD, which has been documented previously[19,23] (Supplementary Fig. 1a). We calculated that the frequency of Leu$^+$ colonies in an un-induced population was about $1.29 \pm 1.04\%$ (Supplementary Fig. 1b). In the presence of β-estradiol, the fraction of Leu$^+$ colonies increased dramatically and reached about 15% in as little as 4 h. Higher frequencies can be achieved by increasing the induction time. As shown in

Supplementary Fig. 1b, approximately 30% of the cells were able to grow on SC-Leu after 24 h' induction. These results suggested that ReSCuES is a sensitive system to measure the activity of Cre recombinase in a cell and can be used as an effective reporter.

Next, ReSCuES was integrated at the *HO* locus of a synXII strain and SCRaMbLE was activated. As shown in Fig. 2a, many Leu+ colonies with large diversity in colony size could be identified after 8-hours' induction. Small colonies greatly outnumbered large ones (Fig. 2a), suggesting that most SCRaMbLE events compromise cell fitness, which is consistent with previous findings[19]. Five colonies of different sizes were randomly isolated and subjected to PCR analysis using a set of selected primer pairs on synXII (Supplementary Data 1). At least one loxPsym site was flanked by each pair of primers, and, therefore, changes in amplicon size could represent that a rearrangement has happened at this position. We found at least one locus with a changed amplicon size in each of these five clones and SCRaMbLE survivor status was further confirmed by whole-genome sequencing results.

### Genome characterization of SCRaMbLEd clones

The genomic DNA of these five clones were extracted and subjected to whole-genome sequencing using the Ion Proton™ platform. The reads were mapped to the synXII reference genome. Splitting reads that did not map to the parental genome were analyzed using Bowtie2 for novel junction identification[25]. Structural variations were reconstructed by combining sequencing depth and split-read mapping, and individually verified by junction PCR (Supplementary Figs. 2–7). All rearrangement types were found, including inversions, deletions and duplications, though only one duplication event was identified (Fig. 2b, c). Gene content and chromosome length of synXII in each SCRaMbLEd strain were primarily unchanged. There were no large deletions or duplications, which would obviously alter the size of synXII (Fig. 2b and Supplementary Figs. 3–7). In agreement with the PCR analysis, we found the presence of unique rearrangements within the synthetic chromosome in each strain, supporting the assumption that random recombinations happen between loxPsym sites during SCRaMbLEing. In addition, we were not able to detect any off-target or ectopic recombinations between loxPsym and cryptic loxP sequences in the genome, indicating the fidelity of SCRaMbLE, consistent with previous report[23]. Furthermore, we found no evidence of rearrangements between loxPsym sites and the loxP sites introduced in ReSCuES, consistent with the orthogonality reported previously[24]. The detailed information about the sequencing results and functional annotations of genes affected by structural rearrangements were summarized in Supplementary Data 2.

Surprisingly, we noticed that the number of rearrangements within each strain is only comparable to that of SCRaMbLEd synIXR strains, despite there being 299 loxPsym sites in synXII, almost ten times more than the number in synIXR. The most recombined strain was JDY509, which included five deletions and seven inversions. three out of the five strains have only three recombination events within the over 2.5 Mb long chromosome (Fig. 2b, c). On the other hand, we found the majority of rearrangements are inversions, which were identified in all strains. In contrast to what reported by Shen et al.[23] that duplications and higher amplifications are predominate after SCRaMbLEing synIXR, we detect only one tandem duplication event among all strains. One potential cause of the difference might be the format and gene content of the chromosome. In their study, a circular chromosome arm was used, whereas here a linear chromosome was SCRaMbLEd. The circular chromosome may favor the SCRaMbLE mediated duplication through a double

rolling circle mechanism[23]. In addition, all events here are simple rearrangements involving one or two recombinations (Supplementary Figs. 3–7). We reasoned that the presence of many essential genes could be an important limiting factor of more complex rearrangements, which result in lethality of the SCRaMbLEd strains. Supporting this hypothesis, we found that six of nine deletions were adjacent to an essential gene and the remaining three were spaced to an essential gene just by one loxPsym site (Supplementary Figs. 3–7). The largest deletion was found in JDY509, removing 13,161 bp of DNA between two essential genes, *BOS1* and *SMC4* (Supplementary Fig. 6c).

The largest structural variant among the five strains was an over 325 kb genome rearrangement resulting from two successive inversions (Fig. 2d), which was confirmed by the split read map and junction PCR (Supplementary Fig. 2a, b). Consequently, the size of the left arm of synXII increased from about 138 kb to over 265 kb (Fig. 2d). Interestingly, despite no coding sequences have been deleted in JDY510, we found significant decrease on growth rate compared to that of the parental strain (Supplementary Fig. 2c, d), suggesting the existence of potential regulatory mechanism by non-coding sequences or higher order chromosome structure.

In summary, whole-genome sequence analysis of the five SCRaMbLEd strains confirmed that SCRaMbLE functions as designed in a strain with a linear synthetic chromosome even longer than 2 Mb. A population of strains with unique rearrangements within the synthetic chromosome can be generated. These strains showed genetic and phenotypical diversity, which can provide valuable resources for further applications.

### Selection of ethanol-tolerant strains generated by SCRaMbLE

In nature, mutations happen spontaneously, leading to harmful, neutral, or advantageous effects. Advantageous mutations are rare and strongly selected for, allowing organisms to adapt to their environment[26]. As a commonly used industrial microorganism for ethanol production, *S. cerevisiae* has been extensively studied to increase its ethanol tolerance to improve titer and productivity[27–29]. Many strategies have been developed including targeted gene deletion or overexpression, global transcription machinery engineering, genome shuffling, and adaptive evolution[30–34]. The implanted SCRaMbLE system in the Sc2.0 project greatly enhances opportunities for genome-wide rearrangements including changes in chromosome contents and structure, producing a pool of new types of mutations and could potentially be a rich source to further increase the ethanol tolerance. We used synXII to demonstrate whether this could be achieved.

We first tested the ethanol tolerance of the synXII strain to identify an appropriate selective condition. SynXII strain carrying a *Pscw11*-Cre-EBD plasmid (*CEN, URA3*) was serially diluted and spotted onto media containing different amount of ethanol. We found no single colony that could grow at the presence of 8% ethanol after 72 h at 30 °C (Supplementary Fig. 8a), suggesting that this concentration is high enough to inhibit cell growth. Therefore, 8% ethanol is used in further experiments. We then tested whether an ethanol-tolerant mutant could be identified after SCRaMbLE without ReSCuES. As shown in Supplementary Fig. 8b, inducing SCRaMbLE obviously produced new strains with increased ethanol tolerance, since cells only grew after induction but not for the group without SCRaMbLE induction. Fifty-three colonies were randomly isolated and subjected to PCR analysis for 91 different loci listed in Supplementary Data 1. We found, surprisingly, that only 13 colonies had changed PCR patterns (summarized in Supplementary Data 1), suggesting that the majority of clones gained ethanol-resistance but not because

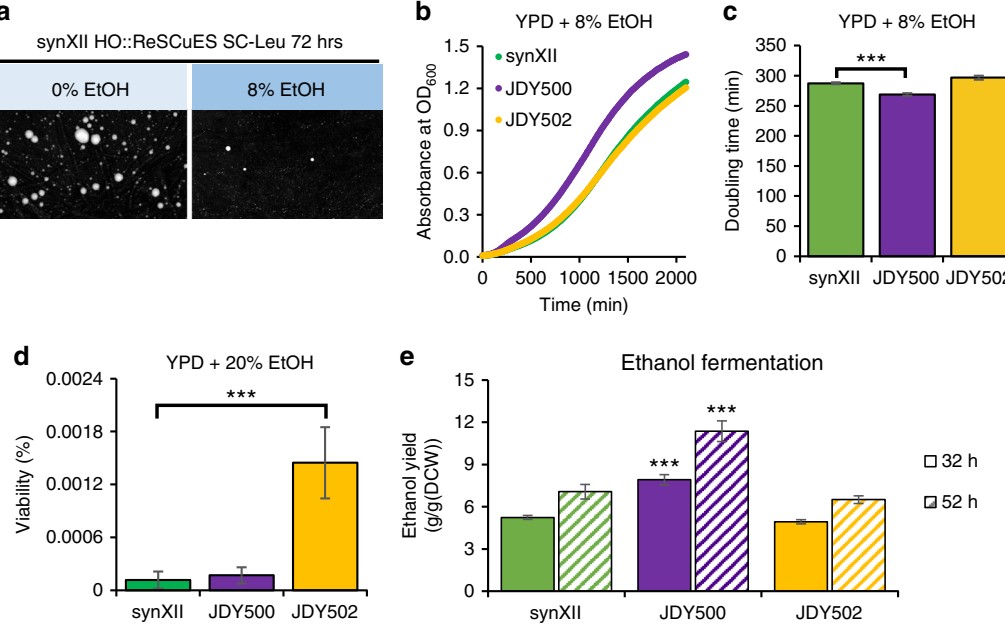

**Fig. 3** Identify the SCRaMbLEd strains with increased ethanol tolerance. **a** Images of clones on SC-Leu plate with (8%) or without (0%) ethanol. The pictures were taken at 72 h after plating. **b** Growth curves of two ethanol-tolerant strains (JDY500 and JDY502) in YPD medium with 8% ethanol compared to that of the original synXII strain. The mean of three biological replicates of each strain was shown. **c** Doubling time (mean ± s.d.) of JDY500 and JDY502 in YPD medium with 8% ethanol compared to that of the original synXII strain. Three biological replicates were measured, *** is for a *P*-value < 0.01 using two-tailed student *t*-test. **d** Viability (mean ± s.d.) of JDY500 and JDY502 compared to that of the original synXII strain after treatment with 20% ethanol for 2 h. Three independent colonies of each strain were tested. *** is for a *P*-value < 0.01 using two-tailed student *t*-test. **e** Measurement of ethanol yields using static fermentation model. The ethanol yield (mean ± s.d.) was measured at 32 and 52 h after switching to the static cultures and presented as gram ethanol per gram dry cell weight (g g$^{-1}$ (DCW)). Three biological replicates were measured, *** is for a *P*-value < 0.01 using two-tailed student *t*-test

of SCRaMbLE. To further confirm this finding, five of the 40 strains without altered PCR patterns were subjected to whole-genome sequencing, which revealed that only one strain, JDY491, contains a small 224 bp deletion (Supplementary Fig. 8c, d). These data strongly suggested that a selection system is necessary to identify "real" SCRaMbLEd cells having a phenotype of interest. We, therefore, applied ReSCuES during SCRaMbLEing. After β-estradiol induction, only several colonies appeared on SC-Leu medium with 8% ethanol, although many colonies could be identified without ethanol (Fig. 3a), consistent with previous data that few SC-Ura with 8% ethanol selected colonies were SCRaMbLEd and suggesting most of the SCRaMbLEd colonies were not ethanol tolerant. Three clones randomly identified from growth on SC-Leu media with 8% ethanol were examined by PCR analysis and whole-genome sequencing, which indicated the presence of rearranged chromosomes in all three strains (Supplementary Fig. 9). Therefore, we conclude that ReSCuES can be applied to identify new strains with rearranged genomes with high efficiency.

Two SCRaMbLEd clones with increased but different ethanol tolerances were profiled in detail (Fig. 3b–e). Cell growth is inhibited at relatively low ethanol concentration and viability is lost at high ethanol concentration, so both faster grow at low ethanol concentration and higher viability at high-ethanol concentration indicate increased ethanol tolerance[35]. One clone, JDY500, grew faster in YPD medium with 8% ethanol with a doubling time significantly shorter than original synXII strain (Fig. 3b, c, mean doubling time shortened by about 19 min). The other clone, JDY502, showed improved viability over that of the original synXII strain under a high-ethanol concentration (Fig. 3d). Interestingly, both JDY500 and JDY502 grew faster than original synXII strain in YPD medium (Supplementary

Fig. 10a). Both strains are genetically stable, as their growth in 8% ethanol was not altered even after passaging over one hundred generations in YPD medium (Supplementary Fig. 10b). Furthermore, the ethanol yields were measured using the static fermentation model established previously[36]. Compared to the synXII strain, the JDY500 strain (with a better growth rate in the presence of ethanol) showed a higher ethanol yield than the JDY502 strain (with a better survival rate in the presence of ethanol) (Fig. 3c–e), which is consistent with previous observations that growth-selected yeast populations have more efficient ethanol fermentation than survival-selected populations[33].

**Dissecting the mechanism of ethanol tolerance.** Ethanol tolerance is a complex polygenic trait. Many efforts have been made to explore underlying mechanisms of this trait[30–38]. Due to the potential multiple changes in a chromosome or a genome, SCRaMbLEd strains with increased ethanol tolerance provide a new platform to dissect mechanisms behind this complex phenotype.

In order to identify the genetic traits related to ethanol tolerance in the isolated SCRaMbLEd strains, we take the advantage of the presence of PCRtags in the synthetic strains, which have previously been used to differentiate synthetic and native DNA[19]. JDY500 was backcrossed with a wild-type strain, followed by sporulation and tetrad dissection. We found most of the tetrads were able to produce four viable spores, which grew similarly in the rich medium (Supplementary Fig. 11a). However, in medium containing ethanol, they could be separated into ethanol-tolerant and -intolerant groups (Fig. 4a and Supplementary Fig. 11b). These two groups of cells were analyzed by PCR, which revealed a region between megachunk M and megachunk

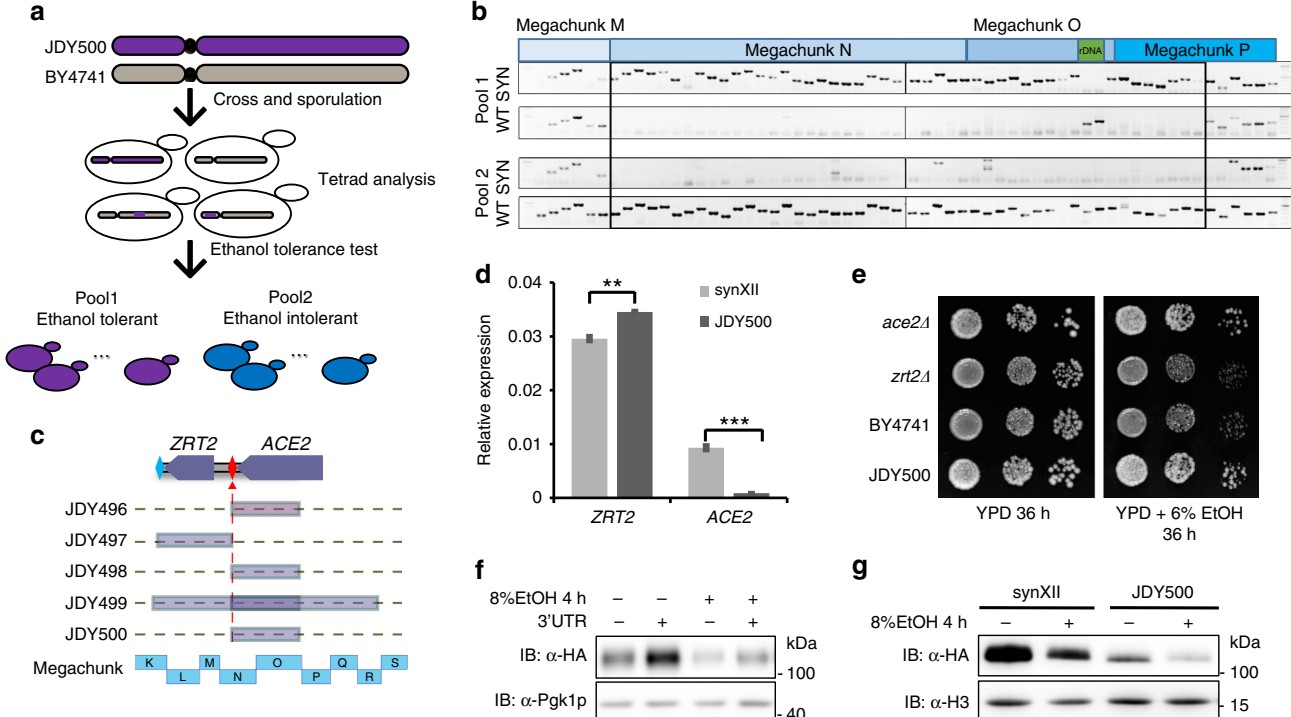

**Fig. 4** Dissect the mechanism of ethanol tolerance. **a** Schematic illustration of procedures to segregate spores from backcross. JDY500 and BY4741 were crossed and sporulated. Tetrads were dissected and tested for ethanol tolerance. The ethanol-tolerant (Pool 1) or intolerant (Pool 2) spores were collected for PCR analysis. **b** PCRTag analysis showed that regions of synthetic chromosome (From megachunk M to megachunk P) are associated with ethanol tolerance. **c** High-throughput sequencing analysis revealed a commonly rearranged junction between *ZRT2* and *ACE2* among five ethanol tolerant strains. The loxPsym sites presented as red diamond between *ZRT2* and *ACE2* was the common loxPsym site used for rearrangements in these five strains. The megachunks were used to indicate the relative position of these rearrangements. **d** Significant decrease of *ACE2* but not *ZRT2* mRNA level in JDY500. Cells at log phase were collected for total mRNA extraction. The detected mRNA level (mean ± s.d.) was normalized to actin. Three biological replicates were measured, ** represents a *P*-value < 0.05 using two-tailed student *t*-test, *** represents a *P*-value < 0.01 using two-tailed student *t*-test. **e** Deletion of *ACE2* but not *ZRT2* generated ethanol tolerant phenotype. Cells at log phase were series diluted and spotted onto YPD medium or YPD medium with 6% ethanol. **f** The 3′UTR is important for Ace2p protein expression. *ACE2* with or without the 3′UTR was cloned into a centromeric plasmid under the control of TEF1 promoter and tagged with 3xHA at N-terminus. The plasmids were transformed into BY4741. Expression of Ace2p was determined by immunoblotting. Pgk1p was used as the loading control. **g** Measure Ace2p expression in JDY500 and synXII. The endogenous Ace2p was tagged with HA and *TEF1* promoter. Expression of Ace2p was determined by immunoblotting. Histone H3 was used as loading control

P for which synthetic PCRtags were only present in ethanol-tolerant cells (Fig. 4b), suggesting that this region might be responsible for ethanol resistance. Similar correlations were also identified by genome sequencing of several other ethanol-tolerant strains with almost identical phenotypes, which were further mapped to rearrangements mediated by the loxPsym site inserted between *ZRT2* and *ACE2* (Fig. 4c and Supplementary Fig. 12). Recombination between this and another loxPsym site could lead to an inversion, potentially causing the loss of function of *ZRT2*, *ACE2*, or both. Quantitative reverse transcription PCR (RT-qPCR) was performed to compare transcription of both genes in synXII and JDY500. The messenger RNA (mRNA) expression of *ZRT2* was only slightly affected in JDY500, whereas that of *ACE2* was almost completely lost compared with synXII (Fig. 4d). In addition, we assayed the ethanol tolerance of strains knocking out either *ACE2 or ZRT2*. *ace2Δ* and JDY500 showed similar levels of ethanol tolerance, but either *zrt2Δ* or *ZRT2* overexpression did not (Fig. 4e and Supplementary Fig. 13). This result is consistent with the previous observation that *ace2Δ* in the diploid genetic background BY4743 is tolerant to ethanol[36]. Altogether, these data strongly argue that loss of *ACE2* leads to increased ethanol tolerance.

Since an inversion in JDY500 removes the native 3′UTR of *ACE2*, a simple hypothesis is that the integrity of the 3′UTR is important for maintaining normal Ace2p level. To test this

hypothesis, a HA-tagged *ACE2* allele with or without 3′UTR was cloned into a centromeric plasmid and introduced into BY4741. Western blots were performed to detect the expression of Ace2p. In the case of the *ACE2* allele without the 3′UTR, expression of Ace2p was significantly decreased (Fig. 4f). Similar results were obtained when cells were treated with ethanol. Furthermore, we could re-capture the ethanol tolerance in the wild-type or synthetic strain by separating the *ACE2* coding sequence from its native 3′UTR using the *URA3* gene (Supplementary Fig. 14). Therefore, we concluded that the reduced Ace2p level in JDY500 is due to the change in its 3′UTR. The expression of Ace2p in JDY500 and synXII was further examined in situ by incorporating an N-terminal 3HA tag under the control of a TEF1 promoter. Subsequent western blot analysis showed that the amount of Ace2p in JDY500 was significantly reduced compared with that in the synXII, irrespective of ethanol treatment (Fig. 4g). Similar results were obtained under its native promoter (Supplementary Fig. 15).

Interestingly, recent work by Wu Y et al.[36] has shown that deleting *ACE2* can greatly increase ethanol yield. Therefore, we examined the ethanol fermentation yields of these SCRaMbLEd strains shown in Fig. 4c. As Expected, the JDY500 strain and additional four SCRaMbLEd strains all showed higher ethanol yields after both 32 and 52 h of fermentation compared with the original synXII strain (Supplementary Fig. 16). This further

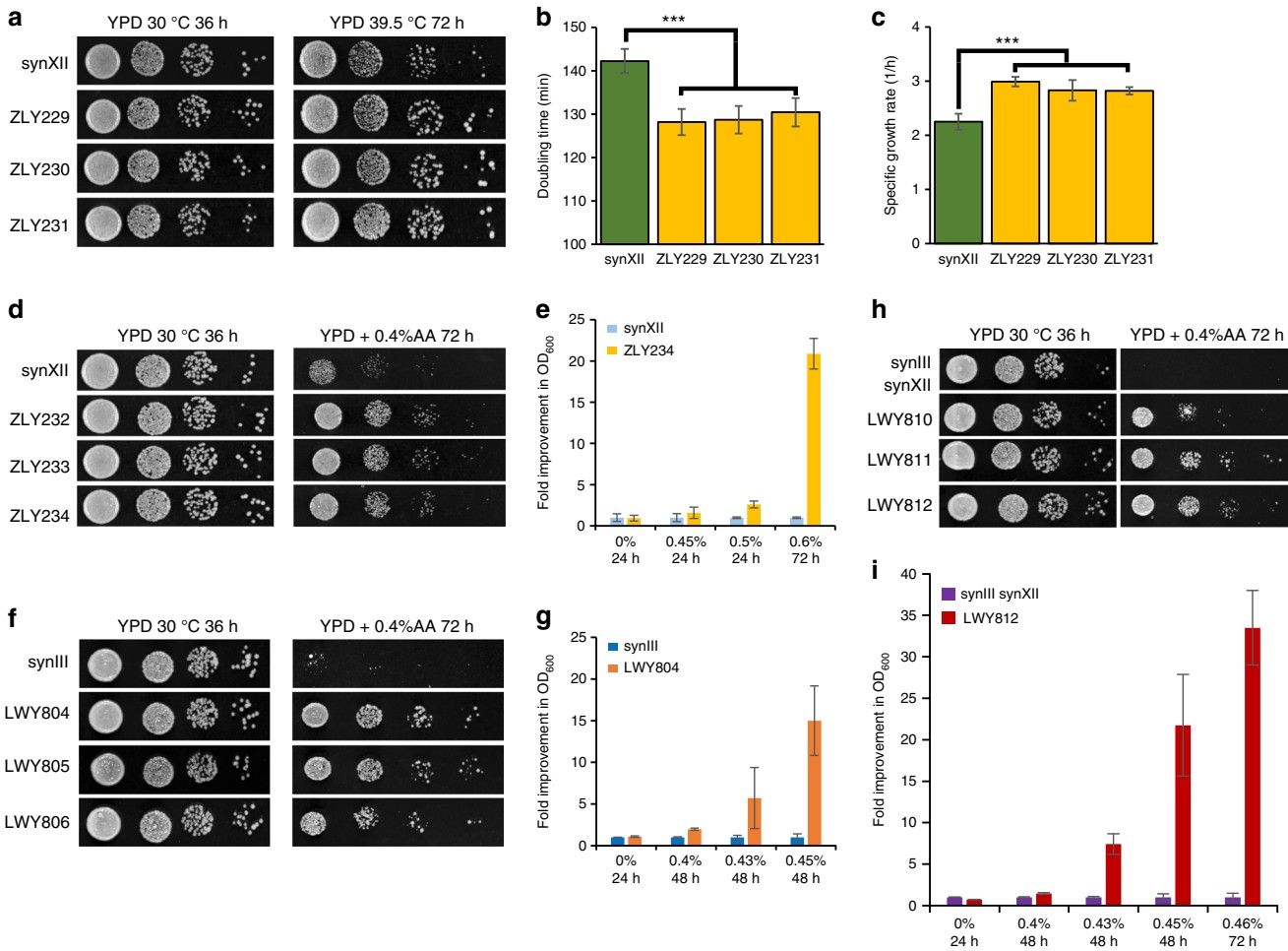

**Fig. 5** Rapid strain improvement by ReSCuES and SCRaMbLE. ReSCuES was applied to identify SCRaMbLEd strains, which are more resistant to higher temperature or acetic acid in strains containing synXII, synIII, or both. Three resistant clones were randomly isolated and tested. **a** SCRaMbLEd synXII strains were more resistant to high temperature as tested by 10× serial dilution. **b** Doubling time (mean ± s.d.) of synXII and the SCRaMbLEd strains in YPD medium under 39.5 °C. Three biological replicates were measured, *** is for a *P*-value < 0.01 using two-tailed student *t*-test. **c** Specific growth rate (h$^{-1}$) (mean ± s.d.) was calculated for synXII and the SCRaMbLEd strains in YPD medium under 39.5 °C. Three biological replicates were measured, *** is for a *P*-value < 0.01 using two-tailed student *t*-test. **d** SCRaMbLEd synXII strains were more resistant to acetic acid (AA) as tested by 10× serial dilution. **e** The growth capacity of ZLY234 was compared to that of the original synXII strain under different level of stress (0%–0.6% acetic acid, V/V%). Three biological replicates were tested for each strain, using an initial OD$_{600}$ = 0.01. The cell density at each time point and acetic acid concentration was normalized to the original synXII strain (mean ± s.d.). **f** SCRaMbLEd synIII strains were more resistant to AA as tested by 10× serial dilution. **g** The growth capacity of LWY804 was compared to that of the original synIII strain under different level of stress (0%–0.45% acetic acid, V/V%). The experiment was performed the same as **e**. **h** SCRaMbLEd synIII,synXII strains were more resistant to AA as tested by 10× serial dilution. **i** The growth of LWY812 was compared to that of the original synIII synXII strain under different level of stress (0–0.46% acetic acid, V/V%). The experiment was performed the same as **e**

supports that disruption of *ACE2* is responsible for the ethanol-related phenotypes of the JDY500 strain.

**General application of ReSCuES for strain engineering.** In addition to identifying strains with increased ethanol tolerance, we applied ReSCuES to select strains with improved fitness under other conditions. Resistance to higher temperature and acetic acid toxicity are two desired features during bio-ethanol fermentation[29,39,40]. Therefore, we tested whether any new strains could be identified after synXII SCRaMbLEing under these two conditions. Similar to results under ethanol selection, some large colonies appeared on a SC-Leu plate at 39.5 °C and a SC-Leu plate with 0.15% acetic acid (Supplementary Fig. 10c). Serial dilution assays with three randomly isolated clones indicated that they were more resistant to these stresses compared with the synXII (Fig. 5a, d). For high-temperature resistant strains, the

doubling time and specific growth rate were measured. As showed in Fig 5b, c, the doubling time of these strains was shortened about 10 min and the specific growth rate was 1.28 ± 0.03 times faster compared to that of synXII. For acetic acid stress, the SCRaMbLEd ZLY234 strain achieved near 21-fold increase in growth capacity compared to synXII strain (Fig. 5e), using a method established before[41]. These results indicated that, combined, ReSCuES and SCRaMbLE can become a powerful tool to quickly generate different strains with improved fitness under various conditions.

All aforementioned SCRaMbLE strains were generated using the synXII strain. We next tested whether ReSCuES can be incorporated into other synthetic yeast strains to produce phenotypic variations. We demonstrated that ReSCuES could be applied in the synIII strain and the synIII synXII strain to identify clones resistant to acetic acid. All three randomly isolated clones outperformed their parental strain in both cases (Fig. 5f,

h). The growth capacity test was applied as before[41] and the SCRaMbLEd LWY812 strain accumulated a near 34-fold more biomass under 0.46% (v/v) acetic acid condition relative to the original synIII synXII strain (Fig. 5g, i). These data suggest that a generic platform has been established for isolating desired advantageous strains with different synthetic genome contents and can be applied to the final Sc2.0 strain.

## Discussion

In this study, ReSCuES was developed to facilitate selection of SCRaMbLEd strains with high efficiency. Although we have demonstrated the system in strains containing one or two synthetic chromosomes, the system will, theoretically, perform even better in strains with more synthetic chromosomes and, therefore, more loxPsym sites incorporated into the genome. As the number of loxPsym sites within the genome increased, the proportion of lethal products of SCRaMbLE increased, making identification of viable SCRaMbLEd clones much harder. This positive selection method could consequently save time and labor compared with previous selection procedures based on smaller colony size or the loss of auxotrophic marker. We have demonstrated that rearranged beneficial mutants, such as those more resistant to high-ethanol concentration, can be isolated and that the underlying mechanism of resistance can be dissected by combining PCR analysis and high-throughput sequencing. Many other applications can be developed, including metabolic chassis engineering and directed population evolution.

ReSCuES was designed based on two nonsymmetrical loxP sites flanking a pair of selectable markers (Fig. 1). Inducing the system results in turning one marker on and turning the other marker off. Therefore, one advantage of this system is that it can be used for multiple rounds to select populations of cells in which recombination has happened. This is extremely useful if SCRaMbLE need to be induced for multiple times, for example, to generate a cell containing a minimal genome.

Similar systems could be constructed on demands using the same design principles, for instance, with fluorescent proteins to replace the auxotrophic markers for flow cytometer analysis. Other derivatives, such as using other inducible promoters to drive Cre expression, could also be applied to increase the frequency of SCRaMbLEd cells in the final viable population. However, when we used Cre under the control of the Gals promoter[42] to mediate SCRaMbLE, we found that controlling the induction time to achieve acceptable viability was not easy, and a strong leaky effect was observed (Supplementary Fig. 17). Additional regulator modules, such as TetO or LacO could be used to combine with the Gal promoter to reduce leaky transcription[43,44].

With the help of ReSCuES, we are able to characterize, for the first time, the SCRaMbLEd linear chromosome in a haploid cell. Several interesting phenomena were discovered. First, in several strains, the location of the recombinant DNA (rDNA) region or centromere was altered. Given that the rDNA region and centromere are two key elements shaping three-dimensional (3D) chromosome structure[45], it will be of interest to find out whether these strains have a different 3D genome structure and whether chromatin structure influences the phenotypes. Second, more inversions than deletions were identified, although the frequencies of each should, in theory, be the same at each loxPsym site. The smallest deletion removed a 314 bp intergenic region (megachunk K in JDY509) and the largest deleted about 13 kb DNA including five non-essential genes (meagchunk J to K in JDY509). The smallest inversion inverted a 531 bp sequence including one gene in megachunk O and the largest inverted nearly one-third of the synthetic chromosome XII. Interestingly, we found many rearranged junctions are near the essential genes,

which suggested that the presence of many essential genes in the linear chromosome might prevent the detection of various deleterious rearrangements in which one or more of these essential genes were removed. Third, inversions are found to be able to fine-tune gene expression and cause phenotypic diversity. For example, JDY510 is the slowest growing strain and contains only inversions. Fourth, although 3'UTR swapping was found to have little effect on fitness by synIXR SCRaMbLE, our results argue that the 3'UTR of *ACE2* is important for its normal mRNA level and ethanol tolerance, indicating that a regulatory function of the 3'UTR in some stress conditions may exist.

Previous work by Shen et al.[23] illustrated that a large diversity of genome arrangements could be detected when SCRaMbLE was induced in a strain containing a synthetic 90-kb circular chromosome arm with 43 recombinase sites. Consistent with their study, we also found that each SCRaMbLEd synXII strain was unique and that rearrangements occurred exclusively at loxPsym sites, even in the presence of loxP sites, demonstrating the capacity of SCRaMbLE. However, to our surprise, much fewer and simpler rearrangements were identified in each SCRaMbLEd strain, including the five randomly selected five Leu+ strains and the ethanol tolerant SCRaMbLEd strains, even though a much longer chromosome with many more loxPsym sites (299 loxPsym sites in synXII compared to 43 loxPsym sites in synIXR) were present and an increased induction time were used. Although the underlying mechanism remains unclear, there are several possibilities: (1) synXII is a linear chromosome. Any rearrangement altering chromosome function will be deleterious. (2) There are many essential segments (the DNA sequence flanked by two loxPsym sites, which includes at least one essential gene or essential chromosome element, for example, centromere) on synXII (about 70 essential segments including 109 essential genes in synXII compared to six essential segments in synIXR including seven essential genes). Many rearrangements lead to loss of viability by deleting an essential segment. (3) synXII harbors long repetitive sequences encoding ribosomal RNA. Rearrangements might be inhibited by the formation of the nucleolus structure.

Several applications of SCRaMbLE with synthetic yeast strains were presented in this study with the ability to quickly select a SCRaMbLEd cell population to, for example, identify strains resistant to various environmental cues. Since SCRaMbLE rearrangements mainly involve large lengths of DNA rather than point mutations, this system could be used in combination with directed evolution to further increase permutations and potentially leads to more of the desired new strains. We have also demonstrated the power of SCRaMbLE not only to identify strains, but also to dissect their potential mechanisms. PCRtag analysis is a quick method to rapidly narrow down the potential positions of the genetic locus. Other commonly used genetics tools such as backcrossing can quickly link phenotype and genotype, avoiding the necessity of sequencing a lot of strains. Many additional applications could be developed based on such synthetic yeast strains in the future.

## Methods

**Integrate the ReSCuES into the HO locus**. The "ReSCuES" cassette used for HO locus integration was PCR amplified from ZLP176 using primer ReSCuES HO F and ReSCuES HO R (Supplementary Data 3). The PCR product was then transformed into BY4741 and synXII using standard lithium acetate transformation method. The Ura+ colonies were selected and verified for the correct integration of ReSCuES.

**Stress tolerance test of the original synXII strain**. To identify a suitable stress condition that could significantly limit the growth of synXII strain but not so harsh to kill the cells, series dilution experiments were conducted. The cells were spotted onto SC-Ura medium with different stresses, such as ethanol and acetic acid. Different concentrations of ethanol or acetic acid were used.

**Induce SCRaMbLE in the synthetic strains**. Synthetic strains were first cultured in medium selected for the Cre plasmid at 30 °C overnight. The cultures were diluted to $OD_{600} = 0.1$ in fresh medium. A proper volume of 1 mM β-estradiol (Sigma E2257, dissolved in 100% ethanol) was added to the final concentration of 1 μM. As a control, equal volume of 100% ethanol was added into another cultures. The growth of cells was monitored by measuring $OD_{600}$ at different time points with a spectrophotometer (Ultrospec 10 cell density meter, Amersham Biosciences). For viability test, cells were collected at 4 and 24 h, and then series dilution was conducted. To measure the colony size, the yeast cells were collected at 0 and 24 h, diluted and plated onto YPD plate to give 100–200 cells per plate. The plates were incubated at 30 °C for 36 h and the colony size was measured using Image J software.

**Select the SCRaMbLEd colonies by ReSCuES**. The synXII strain or BY4741 strain with ReSCuES integrated at HO locus were transformed with the Cre plasmid (ZLP003, *HIS3*). SCRaMbLE was induced in these strains by the addition of 1 μM estradiol as described above. After induction, the cells were plated onto SC-Leu plates with or without stresses. After 3 days, the Leu+ colonies on SC-Leu plates were isolated and streaked for single colonies on the same selective medium for further experiments. For synIII and synIII synXII strains, since there is an essential transfer RNA inserted in the HO locus, plasmid LWP169 containing both *Pscw11*-Cre-EBD and ReSCuES was used.

**Phenotyping of the selected strains**. The selected strains were tested by either serial dilution as describe above or measuring the growth curve in the selected medium. To measure the growth curve, the log phase cells were diluted to $OD_{600} = 0.1$ in a final volume of 100 μl using fresh medium with or without stresses and cultured in the Costar™ Clear Polystyrene 96-well plates. Each strain was measured in triplicates. The 96-well plate was sealed with Breathe-Easy membrane (Sigma, Lot# MKBZ0331) and then cultivated in the Epoch 2 microplate spectrophotometer (Biotek) at 30 °C. $OD_{600}$ was measured every 10 min. Before each measurement, the plate was shaking at 282 cpm for 30 s. The data were analyzed using the on-board Gen5™ software to generate growth curve, doubling time and specific growth rate.

**High-ethanol shock treatment**. The log phase cells were diluted to $OD_{600} = 0.1$ using fresh YPD medium with or without 20% ethanol to a final volume of 5 ml. Samples were taken and labeled as T0. After 2 h, the cells were collected and labeled as T2. Equal volume of sample T0 and T2 were plated onto YPD plates and incubated at 30 °C for 48 h. Viability was calculated by dividing the number of total viable colonies at T2 by that of T0. Three colonies from each strain were tested independently.

**DNA preparation for PCRtag analysis and verification**. Ten OD of yeast cells were collected and washed once with sterile water, resuspended into 100 μl breaking buffer (10 mM Tris-Cl, pH 8.0, 100 mM NaCl, 1 mM EDTA, pH 8.0, 2% (v/v) Triton X-100, 1% (w/v) SDS). Two-hundred microliters of phenol/chloroform/isoamyl alcohol (25:24:1) and approximately 100 μl of 0.5 mm Glass Beads (Biospec, 11079105) were added. The cells were disrupted by Mini-Beadbeater-96 (Biospec, OA60AP-22-1WB) for 2 min. Two-hundred microliters sterile water was added and mixed briefly before centrifugation at 20,238×*g* for 10 min. Two-hundred microliters of top layer was transferred into a new tube containing 500 μl 100% ethanol. DNA was precipitated at −20 °C for 30 min before centrifugation (15,871×*g*, 5 min at 4 °C). The pellet was washed once with 500 μl of 75% ethanol and dried in a vacuum pump (Eppendorf AG 22331 Hamburg, 45 °C, 10 min). The genomic DNA was dissolved in 500 μl of sterile water and stored at −20 °C. Primers designed specifically for junction verification were listed in Supplementary Data 3.

**Structural variant detection with the Ion Proton™ System**. A 200-bp DNA library was prepared for single-end whole-genome sequencing according to the Life Tech standard preparation protocol using the Ion Xpress™ Barcode Adapter 1-96 Kit (Cat.no.4474517) and sequenced on the Ion Proton™ platform. Quality control of sequencing reads was performed. Reads shorter than 30 bp or duplicated were removed. Reads with more than 1% of bases having a Phred-based quality score < 10 or with unknown bases were trimmed first to meet the filtering criteria and removed if the first trimming step failed. Cleaned reads of each sample were mapped to synthetic yeast genome sequences using bowtie2-2.0.0[25] with standard settings. To identify structural variations in each SCRaMbLE genome, we exploit a similar strategy by combining sequencing depth and split-read mapping, to detect complex structural variation and SCRaMbLE events[23]. These reads that did not map to the reference genome were split to pairwise ends (split reads, at least 30 bp) by scanning over all intermediate positions at least 30 bp from the ends of the read. Then these pairwise ends were then aligned to the reference using Bowtie2[25] by single-end mapping with parameter –k 100. The pairwise ends that matched parental sequence were analyzed for breakpoints to provide direct evidence for structural variants. One read was assigned as the most probable mapping type based on five priorities: no recombination events (top priority), intra-chromosome recombination, inter-chromosome recombination within wild-type chromosomes and external chromosome recombination between synthetic and wild-type

chromosomes, and single end mapping. For each identified breakpoint, a 5 bp error range was allowed. Combining the sequencing depth and split-read mapping, the structural variations were reconstructed as described[23].

**RNA extraction**. Cells were cultured overnight in YPD medium at 30 °C with rotation. Then diluted to $OD_{600} = 0.1$, grew to $OD_{600} = 1.0$. 8 OD cells were collected by centrifugation at 4 °C at 845×*g* for 10 min and washed once with iced water, and resuspended in 600 μl TRIzol (Life Technologies, 15596). Three-hundred microliters glass beads were added and the yeast cells were disrupted with Mini-Beadbeater-96 (Biospec, OA60AP-22-1WB) for 1 min. Transferred 600 μl liquid mixture into a new microfuge tube and added 300 μl 1-chloro-3-bromopropane. Vigorously shaked the tube by hand for 3 min. Centrifuged at 13,523×*g* for 6 min, and then transferred upper aqueous layer (~300 μl) into a new tube. RNA was precipitated by adding an equal volume of isopropanol. RNA was pelleted by centrifugation at 13,523×*g* for 10 min at 4 °C. The pellet was washed with 700 μl 75% ethanol and dissolved in 30 μl RNase-free water.

**RT-qPCR**. First-strand cDNA synthesis was performed with 1 μg total RNA of synXII or JDY500, using the FastQuant RT Kit (TIANGEN KR-106-01) with the standard protocol. A 1:5 dilution of the resulting cDNA was used for one RT-qPCR reaction, using the SYBR Green (TIANGEN FP205-02) in 10 μl system. RT-qPCRs were performed in triplicate on the Axgen PCR-96-LP-AB-C Plate using Bio-Rad CFX96 Touch 1 (CFX96 Touch) machine. The amplification conditions were as follows: 95 °C for 15 min, 40 cycles at 95 °C for 10 s, 55 °C for 20 s, and 72 °C 30 s, followed by a melting curve stage. Cycle threshold (CT) values for each pair primers were averaged as the mean ΔCT. The ΔCT of targeted gene divided the ΔCT of the reference gene was used as the relative expression level of targeted gene shown in the figure. Actin was used as the reference gene. The primers used were listed in Supplementary Data 3.

**Analyze Ace2p protein level by western blot**. Ace2p was tagged with 3xHA tags at the N-terminus in synXII or JDY500 strain and the native *ACE2* promoter was replaced with TEF1 promoter simultaneously as described[46]. The strains were cultured overnight, diluted to $OD_{600} = 0.1$, and then grew to $OD_{600} = 0.4$–0.6 in 5 ml YPD medium. Added 100% ethanol to a final concentration of 8% (v/v) or equal volume water as control. After treatment for 4 h, collect 5 OD cells for each sample. Protein samples were prepared using mild alkali treatment and boil method[47]. Five microliters of samples was separated on a 7.5% sodium dodecyl sulfate polyacrylamide electrophoresis gel and transferred to a nitrocellulose membrane for western blot. Mouse monoclonal anti-HA antibody (Sigma, H3663, 1:2000), rabbit polyclonal to anti-H3 antibody (Abcam, ab1791, 1:5000) and rabbit polyclonal to anti-Pgk1p antibody (nordic immununology, NE130/7S, 1:20,000) were used to detect expression of the tagged Ace2p. Uncropped images of all blots appear in Supplementary Fig. 18.

**Data availability**. All data used for this paper are available from the authors on request. Plasmids used in this study have been deposited in Addgene. The sequencing data of the SCRaMbLEd strains have been deposited into NCBI BioSample database with accessions: SAMN07357097-07357116.

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

## Acknowledgements

We are grateful for financial support from National Natural Science Foundation of China (31471254) and the Shenzhen Peacock Team Project (KQTD2015033117210153). This work was also supported by the Biotechnology and Biological Sciences Research Council grant (BB/ M005690/1). We thank Dr Erika Szymanski for proof-reading the manuscript.

## Author contributions

Z.L. and L.W. performed the experiments. Z.L. analyzed data and wrote the manuscript. Y.W. and Y.S. performed the genome sequence analysis of the SCRaMbLEd strains. W.Z. and Y.G. assisted to the construction of the synthetic yeast strains. L.J. contributed to the ethanol fermentation experiment. Q.W. and C.Z. contributed to analysis of the SCRaMbLE data. Y.C. and J.D. wrote the manuscript and supervised the research.

## Additional information

**Competing interests:** The authors declare no competing financial interests.

