## [Peer Review File · Nature Communications]

Reviewers' Comments:

Reviewer #2:

Remarks to the Author:

In the manuscript "Identify and characterize SCRaMbLEd synthetic yeast using ReSCuES" the authors describe a reporter system, 'ReSCuES', that efficiently identifies cells carrying an active Cre recombinase protein. They demonstrate ReSCuES in yeast cells carrying Sc2.0 synthetic yeast chromosomes, which encode 100's of loxPsym sequences as part of the SCRaMbLE system. The major intended use of ReSCuES is to report on Cre activity, whereby activation of the reporter system (a switch between auxotrophic marker expression via Cre-mediated inversion) identifies cells that may have undergone additional recombination events across the synthetic chromosome.

The most broadly interesting aspect of this manuscript to me is the demonstration that a designer, synthetic organism can be inducibly evolved to generate a new phenotype and that the phenotype can be linked back to a specific genotype via mapping/sequencing of the SCRaMbLEd strain. Importantly, the 'evolution' (aka combinatorial synXII rearrangement at loxPsym sites) is achieved in a matter of hours of SCRaMbLE induction, a time-scale that obliterates what could be accomplished in a chemostat or batch culture evolution experiment. This manuscript presents the first clear evidence of SCRaMbLE-dependent gain-of-function with some indication of the underlying mechanism and should be of significant interest to the yeast industrial fermentation and metabolic engineering communities and beyond.

A critical experiment that would unequivocally link Ace2 level to ethanol tolerance is to recapitulate the identified SCRaMbLEd result in an otherwise unmodified synXII and even a wild type strain background. The authors should absolutely perform this experiment.

The authors also present a clear and compelling case that ReSCuES is an important new tool to identify strains carrying synXII SCRaMbLE events. The ReSCuES plasmid should be contributed to Addgene and this information provided in the manuscript.

The authors dilute the impact of the above achievements by presenting a lot of written and experimental documentation of the drawbacks of the SCRaMbLE system (e.g. colony size variation as a means of selection; promoter driving expression of Cre-EBD, MET17 loss for selection of SCRaMbLEd colonies). I strongly suggest that the manuscript be re-structured to place the emphasis on the gain-of-function and phenotype-genotype characterization, identified via the use of ReSCuES. Much less emphasis should be placed on the ins-and-outs of SCRaMbLE, as this is of significantly less interest to the average reader.

In general, I think the manuscript is a real tour-de-force and anyone working with the SCRaMbLE system will appreciate every detail presented. However, in my opinion to appeal to the broader Nature Communications readership requires re-structuring of the manuscript. Further, the manuscript should be very carefully reviewed for sentence structure, word choice, and grammar.

Other comments:

Line 79-81 - The authors suggest a limitation of SCRaMbLE is the daughter specific promoter SCW11. It seems this could easily be overcome by choosing a different yeast promoter, for instance a constitutive promoter. A bigger limitation is probably loss of recombinase activity, which could happen via loss of the Cre plasmid entirely from the cell or by mutation of the recombinase activity, as either would provide a major selective growth advantage. Indeed the authors point this out later in the manuscript (lines 130-132).

Line 105 - PCRTag analysis is not defined.

Line 2 115-117 - The authors should be more specific about this sentence: "During induction,

estradiol binds to EBD and localizes Cre into the nucleus to carry out recombination.” This sentence is not exactly true. The EBD component of Cre-EBD sequesters Cre-EBD in the cytoplasm, mediated via physical interaction with Hsp90. In the presence of estradiol, which binds to the EBD domain and precludes the physical interaction with hsp90, the Cre-EBD protein translocates into the nucleus.

Lines 140- 145 - Colony size variation – I think the main point is that you want to be able to identify colonies of high fitness in which you know at least one SCRaMbLE event has happened. In general, working with small colonies is kind of risky as they are prone to suppressor mutations.

Line 172 – The ReSCUeS plasmid can undergo continued inversion of the Ura3-Leu2 selectable markers. This is an advantage in terms of repeated rounds of SCRaMbLE and selection of colonies in which a Cre recombinase was active. However, if SCRaMbLE is induced for a long period of time, the ReSCUeS reporter could invert multiple times. Do the authors have any idea of the frequency of inversion given time of Cre activity?

Line 307-308 – Only 13/53 colonies had different pcr patterns. How many junctions could you possibly test here compared to how many junctions could have formed? It’s not clear how this is a particularly relevant analysis.

Ace2 & ethanol & SCRaMbLE – an interesting point to make is that the Ace2 protein regulates SCW11 expression. (PMID 11309124). Thus, if reducing Ace2 expression goes hand-in-hand with ethanol tolerance in JDY500, then repeated rounds of SCRaMbLE-ing to further evolve the genotype would likely prove less successful since Cre-EBD presumably would be expressed at a low level. In the long run the Sc2.0 community should move towards an orthogonal induction system for Cre-ebd.

Line 440 – Change sentence to – “Using ReSCuES in synXII cells we have characterized for the first time SCRaMbLEd linear chromosomes in a haploid cell.”

Line 446-447 – is the larger number of inversions compared to deletions statistically significant?

Line 452 – is it statistically significant that rearranged junctions are near essential genes?

Discussion – In the synIXR SCRaMbLE-seq paper, the mean number of SCRaMbLE events was 6.2 ± 4.9 . synIXR had 43 loxP sites and only 7 essential genes. Please provide a more thorough comparison with synXII. For instance, how many non-essential segments are in synXII (i.e. segments that lack essential gene, centromere, telomere etc) and how does this compare to synIXR?

Figure 1 – Clearly label the expected genotypes of the ReSCuES cassette (Ura+/Leu- vs Ura-/Leu+). Define ‘Pura3’. Legend should more clearly state that inverted ReSCuES cassette suggests colonies that may carry additional changes to synthetic chromosomes. Specify the terminators used in the REsCUES cassette somewhere in the manuscript if it is not done so already (either in the diagram, or line 172).

Figure 2 – panel B include coordinates to indicate length of chromosome. Add the approximate position of the rDNA locus. What is an inversion inversion? Consider more descriptive language here, for instance “sequential overlapping inversion”.

Supp Fig 2 – the plate images need to be a lot bigger.

Minor comments

1. Line 43 – change flourishing to wealth

2. line 45 – authors probably meant heterologous rather than heterogeneous.
3. Line 66 – what about duplications? And translocations?
4. Line 70 – change to “Only in the presence of beta-estradiol should the Cre-EBD protein translocate into the nucleus to activate the SCRaMbLE system”.
5. Line 72 – localizes should be localize
6. Line 96 – should be auxotrophic instead of autotrophic
7. Line 96 – change ‘opposite’ to ‘inward pointing’.
8. Line 298 – change selecting to selective
9. Line 422 – change underling to underlying

Reviewer #3:

Remarks to the Author:

The authors use a clever scheme to identify mutants from their SCRaMbLE libraries.

Sequence analysis is conducted to see the type of mutations that arise from this Cre-mediated process. The authors then use this approach to find improvements in phenotype.

Overall, the paper is very superficial in the description and the results are hard to contextualize. I am left wondering how well SCRaMbLE is actually helping the process. No real proper controls are run in this regard.

For example, the authors describe that “majority of clones gained ethanol-resistance but not because of SCRaMbLE”. Certainly there are false positives in any screen. However, there is no comparison of phenotype achieved with and without SCRaMbLE. In particular, what is the benefit achieved with respect to the gained phenotype (like ethanol tolerance) for SCRaMbLE vs other techniques reported in literature as well as simple adaptive evolution. Ethanol tolerance was only slight in 8% ethanol (not a very high level for yeast). Compare with the improvements by Lam et al Science 03 Oct 2014:Vol. 346, Issue 6205, pp. 71-75

Two additional phenotypes were tested (growth at elevated temperature and acetate tolerance).

As for increased temperature, the authors obtain only very slight improvements (not really quantified, only through serial dilution cell blots) at 39.5 C. This is in contrast to significant improvements obtained via adaptive evolution in the paper by Caspeta et al. Science 03 Oct 2014:Vol. 346, Issue 6205, pp. 75-78.

As for increased acetate tolerance, there is again no comparison with state of the art. How does this result and approach compare with Si et al, ACS Syn bio 10.1021/sb500074a where the authors get up to 25 fold increase in growth level for an amount of acetic acid that is >2-fold higher than the value in this paper.

At present, this paper seems to lack the depth that is usually associated with Nature Communications.

Reviewer #4:

Remarks to the Author:

The presented paper reports the development of a reporter system (which they call ReSCuEs for Reporter of SCRaMbLEd Cells using Efficient Selection) allowing researchers to greatly increase the efficiency of the identification of the yeast colonies in which SCRaMbLE (Synthetic Chromosome Rearrangement and Modification by LoxP-mediated Evolution) events took place. This developed strategy will be of great value in the Sc2.0 project, a project that aims to synthesize the complete

S. cerevisiae genome. In Sc2.0, *LoxP* sites, which are necessary for SCRaMbLE, are introduced throughout the yeast genome. Thus, developing ReSCuEs, the way to increase the efficiency of selection for the rearranged strains, is of great importance and is the only way to fully unlock the potential of SCRaMbLE. Application of ReSCuEs will aid in investigating phenomena such as genome minimization, genotype-phenotype associations, and development of improved strains. These last two aspects are included as a proof-of-principle in this study.

Overall, I think the paper is clearly written, provides exciting new data and the developed technology will play a major role in various interesting applications of the synthetic genome developed in the Sc2.0 project. Moreover, the presented case study already provides evidence that the system allows rapid identification of genetic underpinnings of various yeast traits, in this case ethanol tolerance. Therefore, I think this paper is suitable for publications, given that the comments/suggestions listed below are addressed and clarified.

Major comments:

(1) The section from line 120-133 is not clear and should be rephrased. First, from the text it is not obvious that in the first few experiments, the Cre is placed in the HO locus, while for the last experiment it is placed on a plasmid. Moreover, the sentence on line 130 ('...', some have obviously found a way to get rid of the Cre') is too bold, as the authors did not show this at all. If I understand well, they base this claim on an experiment where Cre is on a plasmid, while they should have checked with sequencing of the locus where Cre is introduced whether it is still present or not.

(2) Fig. 5A: the difference in growth rates between the SynXII and "temperature stress resistant" strains is not obvious, while it is claimed in the text a clear difference is observed. Can you provide quantitative data? Raising the temperature from 39.5°C to 41°C can potentially give more clear-cut result.

(3) Line 205-210: It is a pity the authors only showed the frequency of the Rescue switch for two induction times (4 and 24 hours), especially since the induction time selected for future experiments (8 hours) was not included here. A more elaborate investigation of different timepoints would allow a better estimation of the most optimal induction time, without resulting in too much viability loss.

Minor comments:

(1) It would be interesting to include a more thorough analysis of the genes that are deleted/inverted/duplicated in strains JDY506-510. Why exactly do most SCRaMbLEd strains grow slower than the parent on YPD at 30°C? It would be interesting to see what kind of genes were hit (maybe adding a GO table in supplements?).

(2) Line 384: why was 3HA-ACE2 put under TEF1 promoter? Was the expression of this gene under control of the native promoter too low to detect using anti-HA antibodies? Please clarify in the M&M section.

(3) The paper does not contain the analysis of the rearrangements happening in the "temperature stress resistant" (ZLY229-231) and "AcOH stress resistant" (ZLY232-234, LWY804-806, LWY810-812) strains. Were they sequenced? And if so, is it possible to add this information?

(4) Line 134: Maybe it is worth mentioning 'plate effects' as an additional downside of selection for colony size. Colonies which are haphazardly growing right next to other colonies will automatically be smaller compared to colonies growing without direct neighbors.

(5) Line 187-190: this sounds a bit contradictory to data that is presented later on in the text. The authors argue here that since there are hundreds of *loxP* sites, there will automatically be a lot of events when ReSCuEs are activated. Later on, they show that there are only very few events in each mutant. While I still believe their general statement is true (if rescues are activated, there are probably also other events happening), they might want to rephrase this sentence (it is not 'inevitable').

(6) Line 211: It would be nice to check whether longer induction leads to cells with more scramble events. However, we realize that this might be beyond the scope of the presented study.

(7) Line 434-439: please present data (or a reference if available) for this section. The authors claim that putting Cre under the control of the GAL promoter results in leaky expression and huge losses in viability, but it would be nice to support these claims by data.

(8) Line 447: the authors argue that the frequency of inversion and deletions should be similar. However, you have automatically selection against some deletions, because you can't lose essential genes. Therefore, I would argue that the higher incidence of inversions is expected.

(9) Line 459: The authors compare their results with the results obtained by Shen et al. 2016, who observed much more scramble events. One potentially relevant difference between both studies which is not mentioned is the difference in Cre induction time (4 vs 8h). While this likely does not explain the observed difference (because the study with the shorter induction time led to more events..), it might be worth mentioning.

(10) Fig 4C is not very clear, could you explain a bit more elaborate in the legend what the different colors indicate?

(11) Fig 4D: By eye, it looks like ZRT2 expression for the two strains is also significantly different. If so, please indicate on figure.

Typos:

(1) Line 55: the information needs to be updated since according to Richardson et al., Science 355, 1040–1044 (2017), six synthetic chromosomes are completed.

(2) Line 81: "for example survived" should be "for example, survived".

(3) Line 140: add space before 'although'

(4) Line 148: "strain L#1 and L#2" should be "strain S#2 and S#3".

(5) Line 148: replace argued with indicate (data can not 'argue')

(6) Line 167: add 'of' before 'a reporter'

Response to reviewers (NCOMMS-16-30808A)

We thank reviewers for their insightful comments, which will greatly strengthen the manuscript. Please find below detailed responses to each of the concerns raised by reviewers. The corresponding changes are highlighted in the manuscript.

Reviewer #1 Comments to Author

In the manuscript “Identify and characterize SCRaMbLEd synthetic yeast using ReSCuES” the authors describe a reporter system, ‘ReSCuES’, that efficiently identifies cells carrying an active Cre recombinase protein. They demonstrate ReSCuES in yeast cells carrying Sc2.0 synthetic yeast chromosomes, which encode 100’s of loxPsym sequences as part of the SCRaMbLE system. The major intended use of ReSCuES is to report on Cre activity, whereby activation of the reporter system (a switch between auxotrophic marker expression via Cre-mediated inversion) identifies cells that may have undergone additional recombination events across the synthetic chromosome.

The most broadly interesting aspect of this manuscript to me is the demonstration that a designer, synthetic organism can be inducibly evolved to generate a new phenotype and that the phenotype can be linked back to a specific genotype via mapping/sequencing of the SCRaMbLEd strain. Importantly, the ‘evolution’ (aka combinatorial synXII rearrangement at loxPsym sites) is achieved in a matter of hours of SCRaMbLE induction, a time-scale that obliterates what could be accomplished in a chemostat or batch culture evolution experiment. This manuscript presents the first clear evidence of SCRaMbLE-dependent gain-of-function with some indication of the underlying mechanism and should be of significant interest to the yeast industrial fermentation and metabolic engineering communities and beyond.

- A critical experiment that would unequivocally link Ace2 level to ethanol tolerance is to recapitulate the identified SCRaMbLEd result in an otherwise unmodified synXII and even a wild type strain background. The authors should absolutely perform this experiment.

We thank the reviewer for this great suggestion. Additional experiments have been performed and the results were included:

First, we showed that ACE2Δ in the wild type strain background increases the ethanol tolerance to a level similar to that of JDY500 (Fig 4e), strongly suggesting the loss of ACE2 function was the reason for ethanol tolerance in JDY500 (Fig 4d);

Second, we showed that expression of ACE2 will be greatly reduced in the absence of its native 3’UTR (Fig 4f), no matter with or without ethanol treatment. Since an exogenous promoter was used in this assay, we ruled out the possibility that the regulation of ACE2 is from its promoter. Similarly, when we analyzed the protein level in synXII and JDY500, we found Ace2p is reduced in JDY500 (Fig 4 g and supplementary Fig. S14).

Third, we disrupted the regulation by separating the ACE2 coding sequence and its 3'UTR using the URA3 gene, which could recapitulate the identified SCRaMbLEd result in both wild type strain and synXII (Fig S13).

- The authors also present a clear and compelling case that ReSCuES is an important new tool to identify strains carrying synXII SCRaMbLE events. The ReSCuEs plasmid should be contributed to Addgene and this information provided in the manuscript.

We will deposit the plasmid to Addgene as we did before.

- The authors dilute the impact of the above achievements by presenting a lot of written and experimental documentation of the drawbacks of the SCRaMbLE system (e.g. colony size variation as a means of selection; promoter driving expression of Cre-EBD, MET17 loss for selection of SCRaMbLEd colonies). I strongly suggest that the manuscript be re-structured to place the emphasis on the gain-of-function and phenotype-genotype characterization, identified via the use of ReSCuES. Much less emphasis should be placed on the ins-and-outs of SCRaMbLE, as this is of significantly less interest to the average reader.

As suggested, we have removed this part from the manuscript.

- In general, I think the manuscript is a real tour-de-force and anyone working with the SCRaMbLE system will appreciate every detail presented. However, in my opinion to appeal to the broader Nature Communications readership requires re-structuring of the manuscript. Further, the manuscript should be very carefully reviewed for sentence structure, word choice, and grammar.

We have restructured the manuscript accordingly and have it edited by English native speakers.

Other comments:

- Line 79-81 - The authors suggest a limitation of SCRaMbLE is the daughter specific promoter SCW11. It seems this could easily be overcome by choosing a different yeast promoter, for instance a constitutive promoter. A bigger limitation is probably loss of recombinase activity, which could happen via loss of the Cre plasmid entirely from the cell or by mutation of the recombinase activity, as either would provide a major selective growth advantage. Indeed the authors point this out later in the manuscript (lines 130-132).

Yes, we thought initially to replace the SCW11 promoter. We tried to use the Glas controlled Cre-EBD, which failed to achieve acceptable viability due to a strong leaky effect (Discussion part and supplementary Fig. S16). The pSCW11-Cre-EBD offers very tight control.

- Line 105 – PCRTag analysis is not defined.

This has been corrected. The description of PCRTag analysis was included in the

second paragraph of introduction section.

- Line2 115-117 - The authors should be more specific about this sentence: “During induction, estradiol binds to EBD and localizes Cre into the nucleus to carry out recombination.” This sentence is not exactly true. The EBD component of Cre-EBD sequesters Cre-EBD in the cytoplasm, mediated via physical interaction with Hsp90. In the presence of estradiol, which binds to the EBD domain and precludes the physical interaction with hsp90, the Cre-EBD protein translocates into the nucleus.

We thank the reviewer to point this out. According to your suggestion, we have removed this part from the manuscript.

- Lines 140- 145 - Colony size variation – I think the main point is that you want to be able to identify colonies of high fitness in which you know at least one SCRaMbLE event has happened. In general, working with small colonies is kind of risky as they are prone to suppressor mutations.

We agree with the reviewer comments. It is one of the advantage to use the ReSCuES for selection

- Line 172 – The ReSCuES plasmid can undergo continued inversion of the Ura3-Leu2 selectable markers. This is an advantage in terms of repeated rounds of SCRaMbLE and selection of colonies in which a Cre recombinase was active. However, if SCRaMbLE is induced for a long period of time, the ReSCuES reporter could invert multiple times. Do the authors have any idea of the frequency of inversion given time of Cre activity?

This is a good point. We don't know the frequency of inversion given time of Cre activity. The cassette could have been inverted multiple times (1, 3, 5 time for example) in the selected clones as long as it is an odd number. In each round, we will use medium lacking leucine or uracil to select the SCRaMbLEd strains.

- Line 307-308 – Only 13/53 colonies had different pcr patterns. How many junctions could you possibly test here compared to how many junctions could have formed? It's not clear how this is a particularly relevant analysis.

There are 299 *loxPsym* sites in *synXII* and presumably all of them should be checked. In order to reduce the labor, we selected 91 different loci for PCR analysis, which were distributed throughout *synXII*. A clone failed to be detected by the PCR analysis does not necessarily indicate no rearrangements within *synXII*. In contrast, if there is a different PCR patterns, it strongly suggests the presence of a rearrangement event. Initially we thought we should be able to identify many clones with different PCR patterns. This is one of the reason prompting us to develop the ReSCuES system.

- Ace2 & ethanol & SCRaMbLE – an interesting point to make is that the Ace2 protein regulates SCW11 expression. (PMID 11309124). Thus, if reducing Ace2

expression goes hand-in-hand with ethanol tolerance in JDY500, then repeated rounds of SCRaMbLE-ing to further evolve the genotype would likely prove less successful since Cre-EBD presumably would be expressed at a low level. In the long run the Sc2.0 community should move towards an orthogonal induction system for Cre-ebd.

This is a great suggestion and we should find an orthogonal induction system for Cre-EBD. The GAL-LacO, GAL-TetO or a synthetic orthogonal transcription factor might be applied (Ellis et al., 2009, PMID: 19377462; Murphy et al., 2007, PMID: 17652177).

- Line 440 – Change sentence to – “Using ReSCuES in synXII cells we have characterized for the first time SCRaMbLEd linear chromosomes in a haploid cell.”

This has been corrected

- Line 446-447 – is the larger number of inversions compared to deletions statistically significant?

Totally, we identified more inversions (18) than deletions (9). But we can't make a statistical conclusion due to the limited sample number. Using 2-tailed student t-test, the p-value was 0.27. As said by reviewer #3, the higher incidence of inversions may be expected because we could not delete essential genes.

- Line 452 – is it statistically significant that rearranged junctions are near essential genes?

We found that 6 of 9 deletions were adjacent to an essential gene and the remaining 3 were spaced to an essential gene just by one *loxPsym* site. Therefore, we think it should be statistically significant.

- Discussion – In the synIXR SCRaMbLE-seq paper, the mean number of SCRaMbLE events was 6.2 ± 4.9 . synIXR had 43 *loxP* sites and only 7 essential genes. Please provide a more thorough comparison with synXII. For instance, how many non-essential segments are in synXII (i.e. segments that lack essential gene, centromere, telomere etc) and how does this compare to synIXR?

We have included more information about synXII in discussion. In briefly, synXII contains 299 *loxPsym* sites and 109 essential genes. More detailed information about synXII was given in our recent *Science* paper (Zhang et al., 2017, PMID: 28280149).

- Figure 1 –Clearly label the expected genotypes of the ReSCuES cassette (Ura+/Leu- vs Ura-/Leu+). Define ‘Pura3’. Legend should more clearly state that inverted ReSCuES cassette suggests colonies that may carry additional changes to synthetic chromosomes. Specify the terminators used in the REscUES cassette somewhere in the manuscript if it is not done so already (either in the diagram, or line 172).

We have adjusted the labels and re-write the figure legend to include all of these information. We did not use a terminator in the ReSCuES cassette.

- Figure 2 – panel B include coordinates to indicate length of chromosome. Add the approximate position of the rDNA locus. What is an inversion inversion? Consider more descriptive language here, for instance “sequential overlapping inversion”.

We have replaced the “inversion inversion” with “sequential overlapping inversion”, thanks a lot for the suggestion.

The length of chromosome and relative location information were indicated by megachunks. The detailed position information of each structural rearrangements, chromosome length and rDNA region could be found in the corresponding supplemental Fig S3-S7.

- Supp Fig 2 – the plate images need to be a lot bigger.

As suggested, we have removed this part from the manuscript, so the figure was removed, too.

Minor comments

1. Line 43 – change flourishing to wealth

This has been corrected

2. line 45 – authors probably meant heterologous rather than heterogeneous.

This has been corrected

3. Line 66 – what about duplications? And translocations?

Theoretically, both duplications and translocations are possible. However, we only detected one duplication.

4. Line 70 – change to “Only in the presence of beta-estradiol should the Cre-EBD protein translocate into the nucleus to activate the SCRaMbLE system”.

This has been removed

5. Line 72 – localizes should be localize

This has been removed

6. Line 96 – should be auxotrophic instead of autotrophic

This has been corrected

7. Line 96 – change ‘opposite’ to ‘inward pointing’.

This has been corrected

8. Line 298 – change selecting to selective

This has been corrected

9. Line 422 – change underling to underlying

This has been corrected

Reviewer #2 (Remarks to the Author)

The authors use a clever scheme to identify mutants from their SCRaMbLE libraries.

Sequence analysis is conducted to see the type of mutations that arise from this Cre-mediated process. The authors then use this approach to find improvements in phenotype.

Overall, the paper is very superficial in the description and the results are hard to contextualize. I am left wondering how well SCRaMbLE is actually helping the process. No real proper controls are run in this regard.

- For example, the authors describe that “majority of clones gained ethanol-resistance but not because of SCRaMbLE”. Certainly there are false positives in any screen. However, there is no comparison of phenotype achieved with and without SCRaMbLE.

Sorry we didn't make it clear enough. Actually we have compared phenotype achieved with and without SCRaMbLE and the result was shown in Fig S10b. No Colony could appear under the selective condition without SCRaMbLE. We have re-wrote this part in the main text to make it clear

- In particular, what is the benefit achieved with respect to the gained phenotype (like ethanol tolerance) for SCRaMbLE vs other techniques reported in literature as well as simple adaptive evolution.

This is a very good point. However, we are not comparing SCRaMbLE with other techniques. In this manuscript, we are trying to demonstrate for the first time that SCRaMbLE could be used to accelerate genome arrangements for strain improvement. More importantly, we showed that SCRaMbLE can not only identify strains but also help dissecting the underlying mechanism. Certainly, once we have a yeast strain with its entire genome to be synthetic, we can test whether SCRaMbLE could generate a better strain than other techniques. However, we are more intended to use SCRaMbLE as an additional technique for strain improvement. A strategy combining all different techniques should be more desirable.

- Ethanol tolerance was only slight in 8% ethanol (not a very high level for yeast). Compare with the improvements by Lam et al Science 03 Oct 2014:Vol. 346, Issue 6205, pp. 71-75_

Please refer to above response. The purpose of this manuscript is different.

- Two additional phenotypes were tested (growth at elevated temperature and acetate tolerance). As for increased temperature, the authors obtain only very slight improvements (not really quantified, only through serial dilution cell blots) at 39.5 C. This is in contrast to significant improvements obtained via adaptive evolution in the paper by Caspeta et al. Science 03 Oct 2014:Vol. 346, Issue 6205, pp. 75-78.

Please refer to above response. The purpose of this manuscript is different.

- As for increased acetate tolerance, there is again no comparison with state of the art. How does this result and approach compare with Si et al, ACS Syn bio 10.1021/sb500074a where the authors get up to 25 fold increase in growth level for an amount of acetic acid that is >2-fold higher than the value in this paper.

Please refer to above response. The purpose of this manuscript is different.

At present, this paper seems to lack the depth that is usually associated with Nature Communications.

Reviewer #3 (Remarks to the Author)

The presented paper reports the development of a reporter system (which they call ReSCuEs for Reporter of SCRaMbLEd Cells using Efficient Selection) allowing researchers to greatly increase the efficiency of the identification of the yeast colonies in which SCRaMbLE (Synthetic Chromosome Rearrangement and Modification by LoxP-mediated Evolution) events took place. This developed strategy will be of great value in the Sc2.0 project, a project that aims to synthesize the complete *S. cerevisiae* genome. In Sc2.0, LoxP sites, which are necessary for SCRaMbLE, are introduced throughout the yeast genome. Thus, developing ReSCuEs, the way to increase the efficiency of selection for the rearranged strains, is of great importance and is the only way to fully unlock the potential of SCRaMbLE. Application of ReSCuEs will aid in investigating phenomena such as genome minimization, genotype-phenotype associations, and development of improved strains. These last two aspects are included as a proof-of-principle in this study.

Overall, I think the paper is clearly written, provides exciting new data and the developed technology will play a major role in various interesting applications of the synthetic genome developed in the Sc2.0 project. Moreover, the presented case study already provides evidence that the system allows rapid identification of genetic underpinnings of various yeast traits, in this case ethanol tolerance. Therefore, I think this paper is suitable for publications, given that the comments/suggestions listed below are addressed and clarified.

Major comments:

- (1) The section from line 120-133 is not clear and should be rephrased. First, from

the text it is not obvious that in the first few experiments, the Cre is placed in the HO locus, while for the last experiment it is placed on a plasmid. Moreover, the sentence on line 130 ('...', some have obviously found a way to get rid of the Cre') is too bold, as the authors did not show this at all. If I understand well, they base this claim on an experiment where Cre is on a plasmid, while they should have checked with sequencing of the locus where Cre is introduced whether it is still present or not.

Thanks for the comments. We have removed this part of results according to reviewer1's suggestion.

- (2) Fig. 5A: the difference in growth rates between the SynXII and "temperature stress resistant" strains is not obvious, while it is claimed in the text a clear difference is observed. Can you provide quantitative data? Raising the temperature from 39.5°C to 41°C can potentially give more clear-cut result.

We have quantified the growth of these strains. From the results, the doubling time of these strains was shorten about 10 min and the specific growth rate was 1.28 + 0.03 times faster compared to the original SynXII strain (Figure 5b,5c). (No growth at 41°C)

- (3) Line 205-210: It is a pity the authors only showed the frequency of the Rescue switch for two induction times (4 and 24 hours), especially since the induction time selected for future experiments (8 hours) was not included here. A more elaborate investigation of different time points would allow a better estimation of the most optimal induction time, without resulting in too much viability loss.

We have performed experiments with more time points (including the 8 hours as suggested) and updated the figure (Figure S1b).

Minor comments:

- (1) It would be interesting to include a more thorough analysis of the genes that are deleted/inverted/duplicated in strains JDY506-510. Why exactly do most SCRaMbLEd strains grow slower than the parent on YPD at 30°C? It would be interesting to see what kind of genes were hit (maybe adding a GO table in supplements?).

The full list of genes affected by SCRaMbLE mediated genome rearrangements was provided in supplementary table 2 with annotations (sheet 02. annotation). We performed the GO analysis, but unfortunately, no significant enrichments were found. Among the analyzed SCRaMbLEd strains, JDY508 grows comparably with the parent strain (Fig. S2c and S2d), and JDY500 and JDY502 grow faster than the parent strain in YPD medium (Fig S10a). The reviewer raised an interesting point which could be further dissected using ReSCUEs.

- (2) Line 384: why was 3HA-ACE2 put under TEF1 promoter? Was the expression of this gene under control of the native promoter too low to detect using anti-HA

antibodies? Please clarify in the M&M section.

Since we wanted to test the possible effects of ACE2's 3'UTR on its expression, so we chose the N-terminal to add the 3HA tag to avoid possible disruption of 3'UTR's functions. We used a widely used toolbox to add the 3HA tag (Janke et al., 2004, PMID: 15334558). In this toolbox, the N-terminal tagging was associated with the swapping of native promoter to other well-studied promoters, for example, TEF1 promoter that we used. Replacing the native promoter with a well-studied promoter could exclude some potential unknown regulations in the native promoter to make a more persuasive conclusion about 3'UTR's function.

Actually, we have done the western blot under native promoter for SynXII and JDY500. The results were added as Fig S14. The results were similar with that of TEF1 promoter.

M&M section has been rephrased to a more clear description.

- (3) The paper does not contain the analysis of the rearrangements happening in the "temperature stress resistant" (ZLY229-231) and "AcOH stress resistant" (ZLY232-234, LWY804-806, LWY810-812) strains. Were they sequenced? And if so, is it possible to add this information?

These strains were not sequenced.

- (4) Line 134: Maybe it is worth mentioning 'plate effects' as an additional downside of selection for colony size. Colonies which are haphazardly growing right next to other colonies will automatically be smaller compared to colonies growing without direct neighbors.

This has been removed.

- (5) Line 187-190: this sounds a bit contradictory to data that is presented later on in the text. The authors argue here that since there are hundreds of loxPsym sites, there will automatically be a lot of events when ReSCuES are activated. Later on, they show that there are only very few events in each mutant. While I still believe their general statement is true (if rescues are activated, there are probably also other events happening), they might want to rephrase this sentence (it is not 'inevitable').

Thanks, we have rephrased this sentence.

- (6) Line 211: It would be nice to check whether longer induction leads to cells with more scramble events. However, we realize that this might be beyond the scope of the presented study.

Thanks. this is a good hypothesis we will try later

- (7) Line 434-439: please present data (or a reference if available) for this section. The authors claim that putting Cre under the control of the GAL promoter results in leaky expression and huge losses in viability, but it would be nice to support these claims by data.

We have added the data in Fig S16.

- (8) Line 447: the authors argue that the frequency of inversion and deletions should be similar. However, you have automatically selection against some deletions, because you can't lose essential genes. Therefore, I would argue that the higher incidence of inversions is expected.

Yes, we agreed with the reviewer's point and rephrased this sentence to make the description more clear.

- (9) Line 459: The authors compare their results with the results obtained by Shen et al. 2016, who observed much more scramble events. One potentially relevant difference between both studies which is not mentioned is the difference in Cre induction time (4 vs 8h). While this likely does not explain the observed difference (because the study with the shorter induction time led to more events..), it might be worth mentioning.

This has been corrected.

- (10) Fig 4C is not very clear, could you explain a bit more elaborate in the legend what the different colors indicate?

This has been corrected.

- (11) Fig 4D: By eye, it looks like ZRT2 expression for the two strains is also significantly different. If so, please indicate on figure.

Yes, the p-value using two-tailed student t-test was just near the significant level (p=0.048). We added this and a new supplementary figure to illustrate the ZRT2 overexpression is not the reason for ethanol tolerance (Fig S12).

Typos:

- (1) Line 55: the information needs to be updated since according to Richardson et al., Science 355, 1040–1044 (2017), six synthetic chromosomes are completed.

This has been corrected

- (2) Line 81: "for example survived" should be "for example, survived".

This has been corrected.

- (3) Line 140: add space before 'although'

This has been removed.

- (4) Line 148: "strain L#1 and L#2" should be "strain S#2 and S#3".

According to reviewer 1's suggestion, we have removed these result from this manuscript.

- (5) Line 148: replace argued with indicate (data can not 'argue')

This has been removed.

(6) Line 167: add 'of' before 'a reporter'

This has been corrected.

Response to reviewers (NCOMMS-16-30808A)

We thank reviewers for their insightful comments, which will greatly strengthen the manuscript. Please find below detailed responses to each of the concerns raised by reviewers. The corresponding changes are highlighted in the manuscript.

Reviewer #1 Comments to Author

In the manuscript “Identify and characterize SCRaMbLEd synthetic yeast using ReSCuES” the authors describe a reporter system, ‘ReSCuES’, that efficiently identifies cells carrying an active Cre recombinase protein. They demonstrate ReSCuES in yeast cells carrying Sc2.0 synthetic yeast chromosomes, which encode 100’s of loxPsym sequences as part of the SCRaMbLE system. The major intended use of ReSCuES is to report on Cre activity, whereby activation of the reporter system (a switch between auxotrophic marker expression via Cre-mediated inversion) identifies cells that may have undergone additional recombination events across the synthetic chromosome.

The most broadly interesting aspect of this manuscript to me is the demonstration that a designer, synthetic organism can be inducibly evolved to generate a new phenotype and that the phenotype can be linked back to a specific genotype via mapping/sequencing of the SCRaMbLEd strain. Importantly, the ‘evolution’ (aka combinatorial synXII rearrangement at loxPsym sites) is achieved in a matter of hours of SCRaMbLE induction, a time-scale that obliterates what could be accomplished in a chemostat or batch culture evolution experiment. This manuscript presents the first clear evidence of SCRaMbLE-dependent gain-of-function with some indication of the underlying mechanism and should be of significant interest to the yeast industrial fermentation and metabolic engineering communities and beyond.

- A critical experiment that would unequivocally link Ace2 level to ethanol tolerance is to recapitulate the identified SCRaMbLEd result in an otherwise unmodified synXII and even a wild type strain background. The authors should absolutely perform this experiment.

We thank the reviewer for this great suggestion. Additional experiments have been performed and the results were included:

First, we showed that *ACE2Δ* in the wild type strain background increases the ethanol tolerance to a level similar to that of JDY500 (Fig 4e), strongly suggesting the loss of *ACE2* function was the reason for ethanol tolerance in JDY500 (Fig 4d);

Second, we showed that expression of *ACE2* will be greatly reduced in the absence of its native 3’UTR (Fig 4f), no matter with or without ethanol treatment. Since an exogenous promoter was used in this assay, we ruled out the possibility that the regulation of *ACE2* is from its promoter. Similarly, when we analyzed the protein level in synXII and JDY500, we found Ace2p is reduced in JDY500 (Fig 4 g and supplementary Fig. S14).

Third, we disrupted the regulation by separating the *ACE2* coding sequence and its 3'UTR using the *URA3* gene, which could recapitulate the identified SCRaMbLEd result in both wild type strain and *synXII* (Fig S13).

In addition, when our manuscript is in preparation, a paper titled "The transcription factor *Ace2* and its paralog *Swi5* regulate ethanol production during static fermentation through their targets *Cts1* and *Rps4a* in *Saccharomyces cerevisiae*." was published in **FEMS Yeast Research (2016, 16(3). pii: fow022. doi: 10.1093/femsyr/fow022)**. In this paper, the authors screened the yeast deletion collection and found both *ace2Δ* and *swi5Δ* resulted in dramatically increased ethanol yield, probably through the down-regulation of *CTS1* or *RPS4a*. We actually contacted the senior author, Dr. Linghuo Jiang and collaborated on ethanol production during static fermentation using JDY500, which also showed increased ethanol yield (Fig S15).

- The authors also present a clear and compelling case that ReSCuES is an important new tool to identify strains carrying *synXII* SCRaMbLE events. The ReSCuES plasmid should be contributed to Addgene and this information provided in the manuscript.

We will deposit the plasmids to Addgene as we did before.

- The authors dilute the impact of the above achievements by presenting a lot of written and experimental documentation of the drawbacks of the SCRaMbLE system (e.g. colony size variation as a means of selection; promoter driving expression of Cre-EBD, MET17 loss for selection of SCRaMbLEd colonies). I strongly suggest that the manuscript be re-structured to place the emphasis on the gain-of-function and phenotype-genotype characterization, identified via the use of ReSCuES. Much less emphasis should be placed on the ins-and-outs of SCRaMbLE, as this is of significantly less interest to the average reader.

We thank the reviewer for this great suggestion. Therefore, we have removed this part from the manuscript and focused on the development and application of ReSCuES to isolate the gain-of-function mutants and to dissect the underlying mechanism in these mutants using ethanol tolerant as an example.

- In general, I think the manuscript is a real tour-de-force and anyone working with the SCRaMbLE system will appreciate every detail presented. However, in my opinion to appeal to the broader Nature Communications readership requires re-structuring of the manuscript. Further, the manuscript should be very carefully reviewed for sentence structure, word choice, and grammar.

We have restructured the manuscript accordingly and have it edited by English native speakers. We hope the revised version could meet the requirement.

Other comments:

- Line 79-81 - The authors suggest a limitation of SCRaMbLE is the daughter specific promoter SCW11. It seems this could easily be overcome by choosing a different yeast promoter, for instance a constitutive promoter. A bigger limitation is probably loss of recombinase activity, which could happen via loss of the Cre plasmid entirely from the cell or by mutation of the recombinase activity, as either would provide a major selective growth advantage. Indeed the authors point this out later in the manuscript (lines 130-132).

Yes, we thought initially to replace the SCW11 promoter. We tried to use the galactose controlled Cre-EBD, which failed to achieve acceptable viability due to a strong leaky effect (Discussion part and supplementary Fig. S16). The pSCW11-Cre-EBD offers very tight control.

- Line 105 – PCRTag analysis is not defined.

This has been corrected. The description of PCRTag analysis was included in the second paragraph of introduction section.

- Line 2 115-117 - The authors should be more specific about this sentence: “During induction, estradiol binds to EBD and localizes Cre into the nucleus to carry out recombination.” This sentence is not exactly true. The EBD component of Cre-EBD sequesters Cre-EBD in the cytoplasm, mediated via physical interaction with Hsp90. In the presence of estradiol, which binds to the EBD domain and precludes the physical interaction with hsp90, the Cre-EBD protein translocates into the nucleus.

We thank the reviewer to point this out. We have completely revamped the introduction and removed the statement.

- Lines 140- 145 - Colony size variation – I think the main point is that you want to be able to identify colonies of high fitness in which you know at least one SCRaMbLE event has happened. In general, working with small colonies is kind of risky as they are prone to suppressor mutations.

This is correct. It is one of the advantage to use the ReSCuES for selection. The small colonies are much more heterogeneous.

- Line 172 – The ReSCuES plasmid can undergo continued inversion of the Ura3-Leu2 selectable markers. This is an advantage in terms of repeated rounds of SCRaMbLE and selection of colonies in which a Cre recombinase was active. However, if SCRaMbLE is induced for a long period of time, the ReSCuES reporter could invert multiple times. Do the authors have any idea of the frequency of inversion given time of Cre activity?

This is a very interesting question. We don't know the frequency of inversion given time of Cre activity. We assume the cassette could have been inverted multiple times (1, 3, 5 time for example) in the selected clones as long as it is an odd number. In each round, we will use medium lacking leucine or uracil to select the SCRaMbLEd strains.

- Line 307-308 – Only 13/53 colonies had different pcr patterns. How many junctions could you possibly test here compared to how many junctions could have formed? It's not clear how this is a particularly relevant analysis.

There are 299 *loxPsym* sites in *synXII* and presumably all of them should be checked. In order to reduce the labor, we selected 91 different loci for PCR analysis, which were distributed throughout *synXII*. A clone failed to be detected by the PCR analysis does not necessarily indicate no rearrangements within *synXII*. In contrast, if there is a different PCR patterns, it strongly suggests the presence of a rearrangement event. Initially we thought we should be able to identify many clones with different PCR patterns. This is one of the reason prompting us to develop the ReSCuES system.

- Ace2 & ethanol & SCRaMbLE – an interesting point to make is that the Ace2 protein regulates SCW11 expression. (PMID 11309124). Thus, if reducing Ace2 expression goes hand-in-hand with ethanol tolerance in JDY500, then repeated rounds of SCRaMbLE-ing to further evolve the genotype would likely prove less successful since Cre-EBD presumably would be expressed at a low level. In the long run the Sc2.0 community should move towards an orthogonal induction system for Cre-ebd.

Great to know that. We will pay attention to this issue in our future experiments. If this is the case, we should find an orthogonal induction system for Cre-EBD. The GAL-LacO, GAL-TetO or a synthetic orthogonal transcription factor might be applied (Ellis et al., 2009, PMID: 19377462; Murphy et al., 2007, PMID: 17652177).

- Line 440 – Change sentence to – “Using ReSCuES in *synXII* cells we have characterized for the first time SCRaMbLEd linear chromosomes in a haploid cell.”

This has been corrected

- Line 446-447 – is the larger number of inversions compared to deletions statistically significant?

Totally, we identified more inversions (18) than deletions (9). But we can't make a statistical conclusion due to the limited sample number. Using 2-tailed student t-test, the p-value was 0.27. As said by reviewer #3, the higher incidence of inversions may be expected because we could not delete essential genes.

- Line 452 – is it statistically significant that rearranged junctions are near essential genes?

We found that 6 of 9 deletions were adjacent to an essential gene and the remaining 3 were spaced to an essential gene just by one *loxPsym* site. Therefore, we think it should be statistically significant.

- Discussion – In the *synIXR* SCRaMbLE-seq paper, the mean number of SCRaMbLE events was 6.2 ± 4.9 . *synIXR* had 43 *loxP* sites and only 7 essential genes. Please

provide a more thorough comparison with synXII. For instance, how many non-essential segments are in synXII (i.e. segments that lack essential gene, centromere, telomere etc) and how does this compare to synIXR?

We have included more information about synXII in discussion. In briefly, synXII contains 299 *loxPsym* sites and 109 essential genes. More detailed information about synXII was given in our recent *Science* paper (Zhang et al., 2017, PMID: 28280149).

- Figure 1 –Clearly label the expected genotypes of the ReSCuES cassette (Ura+/Leu– vs Ura–/Leu+). Define ‘Pura3’. Legend should more clearly state that inverted ReSCuES cassette suggests colonies that may carry additional changes to synthetic chromosomes. Specify the terminators used in the REsCUes cassette somewhere in the manuscript if it is not done so already (either in the diagram, or line 172).

We have adjusted the labels and re-written the figure legend to include all of the information. We did not use a terminator in the ReSCuES cassette.

- Figure 2 – panel B include coordinates to indicate length of chromosome. Add the approximate position of the rDNA locus. What is an inversion inversion? Consider more descriptive language here, for instance “sequential overlapping inversion”.

We have replaced the “inversion inversion” with “sequential overlapping inversion”, thanks a lot for the suggestion.

The length of chromosome and relative location information were indicated by megachunks. The detailed position information of each structural rearrangements, chromosome length and rDNA region could be found in the corresponding supplemental Fig S3-S7.

- Supp Fig 2 – the plate images need to be a lot bigger.

As suggested, we have removed this part from the manuscript, so the figure was removed, too.

Minor comments

1. Line 43 – change flourishing to wealth

This has been corrected

2. line 45 – authors probably meant heterologous rather than heterogeneous.

This has been corrected

3. Line 66 – what about duplications? And translocations?

Theoretically, both duplications and translocations are possible. However, we only detected one duplication.

4. Line 70 – change to “Only in the presence of beta-estradiol should the Cre-EBD protein translocate into the nucleus to activate the SCRaMbLE system”.

This has been removed

5. Line 72 – localizes should be localize

This has been removed

6. Line 96 – should be auxotrophic instead of autotrophic

This has been corrected

7. Line 96 – change ‘opposite’ to ‘inward pointing’.

This has been corrected

8. Line 298 – change selecting to selective

This has been corrected

9. Line 422 – change underling to underlying

This has been corrected

Reviewer #2 (Remarks to the Author)

The authors use a clever scheme to identify mutants from their SCRaMbLE libraries.

Sequence analysis is conducted to see the type of mutations that arise from this Cre-mediated process. The authors then use this approach to find improvements in phenotype.

Overall, the paper is very superficial in the description and the results are hard to contextualize. I am left wondering how well SCRaMbLE is actually helping the process. No real proper controls are run in this regard.

We thank this reviewer for the critical comments on this manuscript. We have made substantial changes to address each comment and hope the revised manuscript could meet the expectation.

➤ For example, the authors describe that “majority of clones gained ethanol-resistance but not because of SCRaMbLE”. Certainly there are false positives in any screen. However, there is no comparison of phenotype achieved with and without SCRaMbLE.

Sorry we didn’t make it clear enough. The reason why we found “majority of clones gained ethanol-resistance but not because of SCRaMbLE” is because under the selection condition (medium with 8%), resistant clones could arise by different mechanisms such as spontaneous mutation. Only a small portion of these resistant clones come from SCRaMbLE. Without the “ReSCUeS” system, it is very

hard to identify the SCRaMbLEd cells from that of randomly mutated ones. This is the major reason prompting us to develop the selecting system. It will be interesting to compare the phenotypes of the resistant strains generated with or without SCRaMbLE. However, since the yeast strain used in this study contains only a single synthetic chromosome, the result will be highly biased. It will be a great experiment in the future with the yeast strain containing multiple or all synthetic chromosomes.

- In particular, what is the benefit achieved with respect to the gained phenotype (like ethanol tolerance) for SCRaMbLE vs other techniques reported in literature as well as simple adaptive evolution.

This is a very good point and we would love to compare SCRaMbLE with other techniques in literature. However, limited by the progress of the entire Sc2.0 project, we haven't able to generate a strain with its entire genome replaced by the synthetic chromosomes. We can perform the experiments but as stated above, the result will be highly biased if a strain with only one synthetic chromosome was used. Therefore, in this manuscript, we focused on demonstrating the power of SCRaMbLE to generate the genotypic diversity, which could allow us to identify the strains with desired phenotypes. These phenotypes may not be the strongest in the literature since only one chromosome was SCRaMbLEd. Furthermore, it is also possible that we could fail to identify a desired phenotype if the potential causes of this phenotype are not resided in this particular synthetic chromosome. In case of ethanol resistance, we are lucky (and unlucky) that *ACE2*, which was identified recently to be involved in ethanol fermentation, resides on ChrXII. On the other hand, one advantage to use SCRaMbLE is that it is much easier to dissect the underlining mechanism of resistance by PCRTag analysis, as demonstrated in the manuscript.

- Ethanol tolerance was only slight in 8% ethanol (not a very high level for yeast). Compare with the improvements by Lam et al Science 03 Oct 2014:Vol. 346, Issue 6205, pp. 71-75

The improvement in this manuscript is not comparable to that by Lam et al. The reason why we are using much lower ethanol concentration is because we only have strains with a single synthetic chromosome for SCRaMbLE. It will be of great interest to identify strains resistant to much higher ethanol concentration once we have the strain with an entire synthetic genome.

- Two additional phenotypes were tested (growth at elevated temperature and acetate tolerance). As for increased temperature, the authors obtain only very slight improvements (not really quantified, only through serial dilution cell blots) at 39.5 C. This is in contrast to significant improvements obtained via adaptive evolution in the paper by Caspeta et al. Science 03 Oct 2014:Vol. 346, Issue 6205, pp. 75-78.

The improvement in this manuscript is not comparable to that by Caspeta et al with similar reason as stated above.

- As for increased acetate tolerance, there is again no comparison with state of the art. How does this result and approach compare with Si et al, ACS Syn bio 10.1021/sb500074a where the authors get up to 25 fold increase in growth level for an amount of acetic acid that is >2-fold higher than the value in this paper.

The improvement in this manuscript is not comparable to that by Si et al with similar reason as stated above.

At present, this paper seems to lack the depth that is usually associated with Nature Communications.

Reviewer #3 (Remarks to the Author)

The presented paper reports the development of a reporter system (which they call ReSCuEs for Reporter of SCRaMbLEd Cells using Efficient Selection) allowing researchers to greatly increase the efficiency of the identification of the yeast colonies in which SCRaMbLE (Synthetic Chromosome Rearrangement and Modification by LoxP-mediated Evolution) events took place. This developed strategy will be of great value in the Sc2.0 project, a project that aims to synthesize the complete *S. cerevisiae* genome. In Sc2.0, LoxP sites, which are necessary for SCRaMbLE, are introduced throughout the yeast genome. Thus, developing ReSCuEs, the way to increase the efficiency of selection for the rearranged strains, is of great importance and is the only way to fully unlock the potential of SCRaMbLE. Application of ReSCuEs will aid in investigating phenomena such as genome minimization, genotype-phenotype associations, and development of improved strains. These last two aspects are included as a proof-of-principle in this study.

Overall, I think the paper is clearly written, provides exciting new data and the developed technology will play a major role in various interesting applications of the synthetic genome developed in the Sc2.0 project. Moreover, the presented case study already provides evidence that the system allows rapid identification of genetic underpinnings of various yeast traits, in this case ethanol tolerance. Therefore, I think this paper is suitable for publications, given that the comments/suggestions listed below are addressed and clarified.

Major comments:

- (1) The section from line 120-133 is not clear and should be rephrased. First, from the text it is not obvious that in the first few experiments, the Cre is placed in the HO locus, while for the last experiment it is placed on a plasmid. Moreover, the sentence on line 130 ('...', some have obviously found a way to get rid of the Cre') is too bold, as the authors did not show this at all. If I understand well, they base

this claim on an experiment where Cre is on a plasmid, while they should have checked with sequencing of the locus where Cre is introduced whether it is still present or not.

Thanks for the comments. We have removed this part of results according to reviewer1's suggestion. However, to answer the question if Cre is still present or not in a strain containing the Cre integrated at the HO locus, we examined the Nat gene which are linked to the Cre. What we found is that among the SCRaMbLEd clones, there are approximately 5% which are Nat-. We further tested if the Cre is lost or mutated in these Nat- strain and found that 100% of them have lost the Cre. Therefore, upon induction, a small portion(5%) of the cells are able to get rid of the Cre gene.

(2) Fig. 5A: the difference in growth rates between the SynXII and "temperature stress resistant" strains is not obvious, while it is claimed in the text a clear difference is observed. Can you provide quantitative data? Raising the temperature from 39.5°C to 41°C can potentially give more clear-cut result.

We have quantified the growth of these strains. From the results, the doubling time of these strains was shortened about 10 min and the specific growth rate was 1.28 ± 0.03 times faster compared to the original SynXII strain (Figure 5b,5c). In addition, when we increased the temperature to 41°C, the cells failed to grow.

(3) Line 205-210: It is a pity the authors only showed the frequency of the Rescue switch for two induction times (4 and 24 hours), especially since the induction time selected for future experiments (8 hours) was not included here. A more elaborate investigation of different time points would allow a better estimation of the most optimal induction time, without resulting in too much viability loss.

We have performed experiments with more time points (including the 8 hours as suggested) and updated the figure (Figure S1b as shown below).

Fig S1. Verification of ReSCuES in BY4741 background. (a) Cre-EBD generates Leu^+ phenotype in BY4741 strain with ReSCuES integrated at the *HO* locus. A single colony of indicated strains with or without pRS315-*Pscw11*-Cre-EBD plasmid were first overnight cultured in SC-His-Ura medium, and then diluted to $OD_{600}=0.1$ in SC-His medium, at which the time point was set as 0 hr. At each time point, the cells were collected and ten-fold serial dilution was conducted with a start concentration of 0.1 OD/ml. (b) Quantitative analysis the percentage of Leu^+ colonies in (a). Three independent colonies were analyzed for each condition.

Minor comments:

- (1) It would be interesting to include a more thorough analysis of the genes that are deleted/inverted/duplicated in strains JDY506-510. Why exactly do most SCRaMbLEd strains grow slower than the parent on YPD at 30°C? It would be interesting to see what kind of genes were hit (maybe adding a GO table in supplements?).

The full list of genes affected by SCRaMbLE mediated genome rearrangements was provided in supplementary table 2 with annotations (sheet 02. annotation). We performed the GO analysis, but unfortunately, no significant enrichments were found. Among the analyzed SCRaMbLEd strains, JDY508 grows comparably with the parent strain (Fig. S2c and S2d), and JDY500 and JDY502 grow faster than the parent strain in YPD medium (Fig S10a). The reviewer raised an interesting point which could be further dissected using ReSCuES.

- (2) Line 384: why was 3HA-ACE2 put under TEF1 promoter? Was the expression of this gene under control of the native promoter too low to detect using anti-HA antibodies? Please clarify in the M&M section.

Since we wanted to test the possible effects of ACE2's 3'UTR on its expression, so we chose the N-terminal to add the 3HA tag to avoid possible disruption of 3'UTR's functions. We used a widely used toolbox to add the 3HA tag (Janke et al., 2004, PMID: 15334558). In this toolbox, the N-terminal tagging was associated with the swapping of native promoter to other well-studied promoters, for example, TEF1 promoter that we used. Replacing the native promoter with a well-studied promoter could exclude some potential unknown regulations in the native

promoter to make a more persuasive conclusion about 3'UTR's function. Actually, we have done the western blot under native promoter for SynXII and JDY500. The results were added as Fig S14. The results were similar with that of TEF1 promoter.

M&M section has been rephrased to a clearer description.

- (3) The paper does not contain the analysis of the rearrangements happening in the “temperature stress resistant” (ZLY229-231) and “AcOH stress resistant” (ZLY232-234, LWY804-806, LWY810-812) strains. Were they sequenced? And if so, is it possible to add this information?

These strains were not sequenced. We are performing further experiments using strains with multiple synthetic chromosomes, which we think probably will generate much more interested gain-of-function mutants. In this manuscript, we just focused on the development and application of the ReSCuES. We hope this will be okay at current stage because it will take quite some time (also quite some \$\$) to sequence and analyze these strains

- (4) Line 134: Maybe it is worth mentioning ‘plate effects’ as an additional downside of selection for colony size. Colonies which are haphazardly growing right next to other colonies will automatically be smaller compared to colonies growing without direct neighbors.

This has been removed.

- (5) Line 187-190: this sounds a bit contradictory to data that is presented later on in the text. The authors argue here that since there are hundreds of loxPsym sites, there will automatically be a lot of events when ReSCuES are activated. Later on, they show that there are only very few events in each mutant. While I still believe their general statement is true (if rescues are activated, there are probably also other events happening), they might want to rephrase this sentence (it is not ‘inevitable’).

Thanks, we have rephrased this sentence.

- (6) Line 211: It would be nice to check whether longer induction leads to cells with more scramble events. However, we realize that this might be beyond the scope of the presented study.

Thanks. this is a good hypothesis we will try later

- (7) Line 434-439: please present data (or a reference if available) for this section. The authors claim that putting Cre under the control of the GAL promotor results in leaky expression and huge losses in viability, but it would be nice to support these claims by data.

We have added the data in Fig S16.

- (8) Line 447: the authors argue that the frequency of inversion and deletions should

be similar. However, you have automatically selection against some deletions, because you can't lose essential genes. Therefore, I would argue that the higher incidence of inversions is expected.

Yes, we agreed with the reviewer's point and rephrased this sentence to make the description clearer.

(9) Line 459: The authors compare their results with the results obtained by Shen et al. 2016, who observed much more scramble events. One potentially relevant difference between both studies which is not mentioned is the difference in Cre induction time (4 vs 8h). While this likely does not explain the observed difference (because the study with the shorter induction time led to more events..), it might be worth mentioning.

This has been corrected.

(10) Fig 4C is not very clear, could you explain a bit more elaborate in the legend what the different colors indicate?

This has been corrected.

(11) Fig 4D: By eye, it looks like ZRT2 expression for the two strains is also significantly different. If so, please indicate on figure.

Yes, the p-value using two-tailed student t-test was just near the significant level ($p=0.048$). We added this and a new supplementary figure to illustrate the ZRT2 overexpression is not the reason for ethanol tolerance (Fig S12).

Typos:

(1) Line 55: the information needs to be updated since according to Richardson et al., Science 355, 1040–1044 (2017), six synthetic chromosomes are completed.

This has been corrected

(2) Line 81: "for example survived" should be "for example, survived".

This has been corrected.

(3) Line 140: add space before 'although'

This has been removed.

(4) Line 148: "strain L#1 and L#2" should be "strain S#2 and S#3".

According to reviewer 1's suggestion, we have removed these result from this manuscript.

(5) Line 148: replace argued with indicate (data can not 'argue')

This has been removed.

(6) Line 167: add 'of' before 'a reporter'

This has been corrected.

Reviewers' Comments:

Reviewer #2:

Remarks to the Author:

Re-review

The revised manuscript addresses all of my concerns and I have only minor comments as noted below. I support publication of this manuscript.

Analysis of SCRaMbLEd synXII genomes (JDy506-510): It is possible that the low yield of SCRaMbLE events is a result of sequencing only 5 strains; Shen et al sequenced >50 strains. However, it is more likely that by selecting high fitness SCRaMbLEd colonies the authors have effectively filtered for fewer recombination events; with so many essential segments, it seems likely that lots of rearrangement would reduce fitness and/or be lethal. While not necessary here, the authors could try SCRaMbLEing in a heterozygous diploid to compensate for deletion of essential segments and then quantify the number of SCRaMbLE events by sequencing the derivative strains.

Line 229 – I disagree that SCRaMbLE enhances opportunity to find changes in chromosome number and ploidy.

Backcrossing SCRaMbLEd synXII to wildtype: This is a good strategy to map a phenotype to a specific SCRaMbLE event. The major limitation here is if the SCRaMbLEd synthetic chromosome is so different from the unSCRaMbLEd version that the heterozygous diploid will not go through meiosis efficiently. The authors should include information about the formation of 4 spore tetrads in text and supplemental information related to figure 4.

Line 428 – There are many times when >1 essential gene is encompassed between two loxPsym sites on synXII. I suggest the authors report the number of essential segments rather than the number of essential genes. There are probably about 70 essential segments in synXII (including the centromere and telomere). It is misleading to report the total number of essential genes. It would be more relevant to report the fraction of essential segments for both synIXR and synXII.

Minor corrections:

Figure 3b – typo in the word absorbance

Reviewer #3:

Remarks to the Author:

The authors have addressed most comments in the last review and this manuscript should now be acceptable for publication.

Reviewer #4:

Remarks to the Author:

I am delighted to see that the authors have addressed all remarks. I think that the extra experiments and textual changes further strengthen what was essentially already a solid and interesting report of an exciting new technique, and I strongly support publication of this revised version. I personally also support the notion that this is essentially a report of a new technology, and that the applications and comparison to natural evolution, and other techniques to increase the rate of adaptive evolution, is left for follow-up studies.

Respond to reviewers

Please find the point-by-point responses to the reviewer's comments below which are marked in red.

REVIEWERS' COMMENTS:

Reviewer #2 (Remarks to the Author):

Re-review

The revised manuscript addresses all of my concerns and I have only minor comments as noted below. I support publication of this manuscript.

Analysis of SCRaMbLEd *synXII* genomes (JDy506-510): It is possible that the low yield of SCRaMbLE events is a result of sequencing only 5 strains; Shen et al sequenced >50 strains. However, it is more likely that by selecting high fitness SCRaMbLEd colonies the authors have effectively filtered for fewer recombination events; with so many essential segments, it seems likely that lots of rearrangement would reduce fitness and/or be lethal. While not necessary here, the authors could try SCRaMbLEing in a heterozygous diploid to compensate for deletion of essential segments and then quantify the number of SCRaMbLE events by sequencing the derivative strains.

This is a very good point. SCRaMbLEing in a heterozygous diploid is a good idea and could be pursued later.

Line 229 – I disagree that SCRaMbLE enhances opportunity to find changes in chromosome number and ploidy. We have changed the statement to remove chromosome number and ploidy.

Backcrossing SCRaMbLEd *synXII* to wildtype: This is a good strategy to map a phenotype to a specific SCRaMbLE event. The major limitation here is if the SCRaMbLEd synthetic chromosome is so different from the unSCRaMbLEd version that the heterozygous diploid will not go through meiosis efficiently. The authors should include information about the formation of 4 spore tetrads in text and supplemental information related to figure 4. We agree with the reviewer that the major limitation of our mapping strategy is meiosis efficiency. It is possible that the diploid won't be able to sporulate appropriately. Luckily, in our case, we found most of the tetrads were able to form four viable spores, despite the growth differences. The tetrad analysis is shown below and also included into the supplemental information.

We added the following statement into the text: "JDY500 was backcrossed with a wild type strain, followed by sporulation and tetrad dissection. We found most of the tetrads were able to produce four viable spores which grew similarly in the rich medium (Supplementary Fig. S11). However, in medium containing ethanol, they could be separated into ethanol-tolerant and -intolerant groups (Fig. 4a and Supplementary Fig. S11)"

Line 428 – There are many times when >1 essential gene is encompassed between two *loxP* sites on *synXII*. I suggest the authors report the number of essential segments rather than the number of essential genes. There are probably about 70 essential segments in *synXII* (including the centromere and telomere). It is misleading to report the total number of essential genes. It would be more relevant to report the fraction of essential segments for both *synIXR* and *synXII*.

This is a good idea. We have replaced the number of essential genes by essential segments as suggested. There are about 70 essential segments (>=1 essential gene encompassed between two *loxP* sites) in *synXII* vs 6 essential segments in *synIXR*.

Minor corrections:

Figure 3b – typo in the word absorbance

This has been corrected

Reviewer #3 (Remarks to the Author):

The authors have addressed most comments in the last review and this manuscript should now be acceptable for publication.

Reviewer #4 (Remarks to the Author):

I am delighted to see that the authors have addressed all remarks. I think that the extra experiments and textual changes further strengthen what was essentially already a solid and interesting report of an exciting new technique, and I strongly support publication of this revised version. I personally also support the notion that this is essentially a report of a new technology, and that the applications and comparison to natural evolution, and other techniques to increase the rate of adaptive evolution, is left for follow-up studies.